# HOW TRANSFORMERS GET RICH: APPROXIMATION AND DYNAMICS ANALYSIS

## ABSTRACT

Transformers have demonstrated exceptional in-context learning capabilities, yet the theoretical understanding of the underlying mechanisms remains limited. A recent work (Elhage et al., 2021) identified a "rich" in-context mechanism known as induction head, contrasting with "lazy" $n$-gram models that overlook long-range dependencies. In this work, we provide both approximation and dynamics analyses of how transformers implement induction heads. In the *approximation* analysis, we formalize both standard and generalized induction head mechanisms, and examine how transformers can efficiently implement them, with an emphasis on the distinct role of each transformer submodule. For the *dynamics* analysis, we study the training dynamics on a synthetic mixed target, composed of a 4-gram and an in-context 2-gram component. This controlled setting allows us to precisely characterize the entire training process and uncover an *abrupt transition* from lazy (4-gram) to rich (induction head) mechanisms as training progresses. The theoretical insights are validated experimentally in both synthetic and real-world settings.

## 1 INTRODUCTION

Transformer, introduced by Vaswani et al. (2017), have achieved remarkable success across various domains, including natural language processing, computer vision, and scientific computing. An emergent observation is that transformers, trained on trillions of tokens, can perform (few-shot) in-context learning (ICL), which makes prediction based on the contextual information without needing model retraining (Brown et al., 2020). This ICL ability is widely regarded as crucial for enabling large language models (LLMs) to solve reasoning tasks, representing a key step toward more advanced artificial intelligence.

To understand how transformers implement ICL, Elhage et al. (2021) and Olsson et al. (2022) identified a simple yet powerful mechanism known as **induction head**. Specifically, given an input sequence `[···ab···a]`, an induction head predicts `b` as the next token by leveraging the prior occurrence of the pattern `ab` in the context, effectively modeling an in-context bi-gram. In contrast, traditional $n$-gram model (Shannon, 1948) (with a small $n$) utilizes only a limited number of recent tokens to predict the next token, which is context-independent and inevitably overlooks long-range dependence. Based on the extent of context utilization, we categorize $n$-gram model as a *"lazy" mechanism*, whereas the induction head represents a more **"rich" mechanism**.

Practically, induction heads have been demonstrated to play a critical role in enabling LLMs' ICL capabilities (Song et al., 2024; Crosbie and Shutova, 2024), and even used to test new LLM architectures (Gu and Dao, 2023). Theoretically, induction heads also serve as a controllable tool for understanding various aspects of LLMs, such as multi-step reasoning (Sanford et al., 2024b) and inductive biases of different architectures (Jelassi et al., 2024).

In this paper, we aim to provide a theoretical analysis of how transformers can efficiently implement induction heads. The first key problem is to rigorously formalize induction heads and evaluate the efficiency of transformers in representing them. According to Elhage et al. (2021), the original induction head can be implemented using a two-layer, twelve-head transformer without feed-forward networks (FFNs). However, practical scenarios demand more powerful induction heads. Thus, it is crucial to generalize the mechanism behind and explore how different transformer submodules, such as varying the number of attention heads or incorporating FFNs, impact the transformer's ability to implement them. This forms our first research objective:

*(Approximation). Investigate how two-layer transformers express the induction head mechanism and its potential variants.*

The next problem is to investigate the dynamics of transformers in learning induction heads. The pioneering works by Elhage et al. (2021) and Olsson et al. (2022) demonstrated that transformers undergo an abrupt phase transition to learning induction heads. A recent empirical study on synthetic datasets replicate this behavior, further showing that 2-gram is always learned prior to induction heads (Bietti et al., 2024). However, a rigorous theoretical analysis of this learning progression is still lacking. Closing this gap forms our second research objective:

*(Dynamics). Understand how transformers transition from relying on $n$-gram patterns to employing the induction head mechanism as training progresses.*

Focusing on these two key problems, in this paper, we make the following contributions:

- **Approximation analysis: how transformers express induction heads.** We consider three types of induction heads with varying complexities. First, we show that two-layer, single-head transformers without FFNs can efficiently approximate the vanilla induction head (Elhage et al., 2021). We then introduce two generalized induction heads, which leverage richer in-context $n$-gram information and incorporate a general similarity function. Our analysis clarifies the distinct roles of multihead attention, positional encoding, dot-product structure, and FFNs in implementing these generalized induction heads.

- **Dynamics analysis: how learning undergoes a sharp transition from $n$-gram to induction head.** We study the learning dynamics of a two-layer transformer without FFNs for a mixed target, composed of a 4-gram and an in-context 2-gram component. This toy setting allows us to capture the entire training process precisely. Specifically, we show that learning progresses through four phases: partial learning of the 4-gram, plateau of induction head learning, emergence of the induction head, and final convergence, showcasing a sharp transition from 4-gram to induction head. Our analysis identifies two key drivers of the transition: 1) time-scale separation due to low- and high-order parameter dependencies in self-attention, and 2) speed differences caused by the relative proportions of the two components in the mixed target.

Finally, we conduct a series of experiments, ranging from simple toy models to real-world natural language training tasks, to validate our theoretical insights.

Due to space limitations, the detailed discussion of related works is deferred to Appendix A.

## 2 PRELIMINARIES

**Notations.** For $k \in \mathbb{N}^+$, let $[k] = \{1, 2, \ldots, k\}$. For a vector $v$ and $1 \leqslant p \leqslant \infty$, we denote by $\|v\|_p$ the $\ell_p$ norm of $v$. For a matrix $A = (a_{i,j})$, we denote by $\|A\|$, $\|A\|_F$ the spectral and Frobenius norms, respectively; let $\|A\|_{1,1} = \sum_{i,j} |a_{i,j}|$. For an event $S$, we define $\mathbb{I}\{S\} = 1$ if $S$ is true, and $0$ otherwise. We use $\mathrm{sm}(\cdot)$ to denote the softmax function. We use standard big-O notations $\mathcal{O}, \Omega, \Theta$ to hide absolute positive constants, and use $\tilde{\mathcal{O}}, \tilde{\Omega}, \tilde{\Theta}$ to further hide logarithmic constants.

**Sequence modeling.** Given a sequence of tokens $(x_1, x_2, x_3, \ldots)$ with each token lying in $\mathbb{R}^d$, let $X_L = (x_1, x_2, \ldots, x_L) \in \mathbb{R}^{d \times L}$ and $X_{m:n} = (x_m^\top, x_{m+1}^\top, \ldots, x_n^\top)^\top \in \mathbb{R}^{(n-m+1)d}$. Given $A = (a_1, \cdots a_n) \in \mathbb{R}^{m \times n}$, we denote $(a_s)_{s=i}^j = (a_i, \cdots, a_j) \in \mathbb{R}^{m \times (j-i+1)}$. Then, we consider the next-token prediction task: predict $x_{L+1}$ using $X_L = (x_1, x_2, \ldots, x_L)$.

In a $n$**-gram model** (Shannon, 1948), the conditional probability of predicting the next token is given by $p(x_{L+1}|X_L) = p(x_{L+1}|X_{L-n+2:L})$, meaning that the prediction depends only on the most recent $n-1$ tokens. In practice, the value of $n$ is typically small (e.g., 2, 3, or 4), as the computational cost of $n$-gram models grows exponentially with $n$. However, $n$-gram models with small $n$ cannot capture long-range interactions, leading to inferior performance in sequence modeling.

**Transformer** is designed to more efficiently capture long-range dependencies in sequence modeling (Vaswani et al., 2017). Specifically, given an $L$-token input sequence $X = (x_1, \cdots, x_L) \in \mathbb{R}^{d \times L}$, an $U$-layer transformer TF processes it as follows. First, each input token is embedded into a higher-dimensional space through an *embedding layer*: $x_s^{(0)} = W_E x_s + b_E$, $s \in [L]$, where $W_E \in \mathbb{R}^{D \times d}, b_E \in \mathbb{R}^D$.

Next, the $U$-layer SA-FFN blocks process the embedded sequence $X^{(0)} = (x_1^{(0)}, \cdots, x_L^{(0)})$ as follows, and the output of the final layer is taken as the output sequence $\mathsf{TF}(X) = X^{(U)} \in \mathbb{R}^{D \times L}$:

$$X^{(u)} = X^{(u-\frac{1}{2})} + \mathsf{FFN}^{(u)}(X^{(u-\frac{1}{2})}), \quad X^{(u-\frac{1}{2})} = X^{(u-1)} + \mathsf{SA}^{(u)}(X^{(u-1)}), \ u \in [U]. \quad (1)$$

Here, $\mathsf{FFN}^{(u)}$ denotes a (token-wise) two-layer FFN of width $M$, and $\mathsf{SA}^{(u)}$ represents the $H$-head self-attention operation. Specifically, when applied to a sequence $Z = (z_1, \cdots, z_L) \in \mathbb{R}^{D \times L}$, $\mathsf{SA}^{(u)}$ operates it as: $\mathsf{SA}^{(u)}(Z) = W_O^{(u)} \sum_{h=1}^{H} \mathsf{SA}^{(u,h)}(Z)$, where

$$\mathsf{SA}^{(u,h)}(Z) = \left(W_V^{(u,h)} Z\right) \mathrm{softmax}\left(\left\langle W_Q^{(u,h)} Z, W_K^{(u,h)} Z\right\rangle + R^{(u,h)}\right), \quad (2)$$

where $W_Q^{(u,h)}, W_K^{(u,h)}, W_V^{(u,h)}, W_O^{(u)} \in \mathbb{R}^{D \times D}$ correspond to the query, key, value and output matrices of the $(u, h)$-th head, respectively. $\mathrm{softmax}$ represents taking softmax normalization across columns. $\left\langle W_Q^{(u,h)} X, W_K^{(u,h)} X\right\rangle$ is called the dot-product (DP) structure. Furthermore, $R^{(u,h)} = (R_{i,j}^{(u,h)}) \in \mathbb{R}^{L \times L}$ denotes the additive relative positional encoding matrix, which satisfies $R_{i,j}^{(u,h)} = -\infty$ if $i \leqslant j$ for the next-token prediction task.

**Relative positional encoding** (RPE). Throughout this paper, we focus on the Alibi RPE (Press et al., 2022), where $R_{ij}^{(u,h)}$ exhibit a Toeplitz structure, i.e., $R_{ij}^{(u,h)} = \phi(i - j; p^{(u,h)})$ for $i, j \in [L]$. Here, $p^{(u,h)}$'s are learnable parameters and $\phi(\cdot; p)$ has the form: $\phi(z; p) = \begin{cases} -p \cdot (z - 1), & \text{if } z \geqslant 1 \\ -\infty, & \text{otherwise} \end{cases}$. Note that we adopt the Alibi only for simplicity and our results can be extended to other additive RPEs, such as T5 (Raffel et al., 2020) and KERPLE (Chi et al., 2022). However, extending our analysis to the popular rotary RPE (Su et al., 2024) may be nontrivial, and we leave this for future work.

## 3 FORMULATION AND APPROXIMATION OF INDUCTION HEAD

In this section, we formalize three types of induction head mechanisms with varying levels of complexity. We then theoretically investigate how two-layer single- or multi-head transformers, with or without FFNs, can efficiently implement these mechanisms, highlighting the distinct roles of different transformer submodules

### 3.1 VANILLA INDUCTION HEADS

The original induction head, proposed in Elhage et al. (2021) and Olsson et al. (2022), is regarded as one of the key mechanisms to implement ICL and reasoning. This induction head suggests that two-layer multi-head transformers without FFNs can execute a simple in-context algorithm to predict the next token b from a context [···ab···a] through retrieval, copying, and pasting, based on in-context bi-gram pairs, as illustrated in Figure 1.

Figure 1: An illustration of the original induction head (taken from Elhage et al. (2021)). The induction head proceeds the context [··· The D] by retrieving the preceding information most relevant to the current token (D), then copying and pasting the subsequent token (the green urs) as the current prediction. Notably, the first and second self-attention layers focus on the highlighted red and green tokens, respectively. For further details, refer to the description below Theorem 3.1.

**Formulation of $\mathsf{IH}_2$.** Based on the phenomenon illustrated in Figure 1, we define the vanilla induction head $\mathsf{IH}_2 : \cup_{L \in \mathbb{N}^+} \mathbb{R}^{d \times L} \mapsto \mathbb{R}^d$ as follows:

$$\mathsf{IH}_2(X_L) = \sum_{s=2}^{L-1} x_s \, \mathrm{sm}\left(\left(x_L^\top W^\star x_{\nu-1}\right)_{\nu=2}^{L-1}\right)_{\nu=s}. \quad (3)$$

Specifically, $\mathsf{IH}_2$ retrieves in-context information based on the similarities of in-context bi-gram pairs $\{(x_s, x_L)\}_{s=1}^{L-2}$. Note that the magnitude of matrix $W^\star$ controls the sparsity of retrieval, since increasing $\|W^\star\|$ causes the softmax output to concentrate as a delta measure over the preceding tokens. Additionally, $\mathsf{IH}_2$ can handle input sequences of arbitrary length.

This model retrieves previous tokens $x_{s-1}$'s that are similar to the current token $x_L$ based on a dot-product similarity, and then copies and pastes $x_{s-1}$'s subsequent token $x_s$ as the current prediction $x_{L+1}$. For example, in Figure 1, the current token $x_L$ is D, and the model retrieves previous tokens similar to D, copying and pasting its subsequent token `urs` as the prediction.

**Comparison with previous formulations.** As shown in Figure 1, the current token D appears multiple times in the preceding context, and the induction head detects all occurrences of D. Our formulation (3) captures this behavior, as the softmax scores for all preceding D are identical. In contrast, previous formulations, such as Sanford et al. (2024a) and Sanford et al. (2024b), focus solely on the most recent occurrence of D, neglecting this multi-occurrence aspect.

**Measure of approximation.** Consider a target function $\mathsf{H} : \cup_{L\in\mathbb{N}^+}\mathbb{R}^{d\times L} \mapsto \mathbb{R}^d$, where $d$ is the token dimension and $L$ denotes the sequence length. Given an input sequence $X \in \mathbb{R}^{d\times L}$, transformer $\mathsf{TF}$ approximates $\mathsf{H}(X)$ using its last output token, i.e., $\mathsf{TF}_{-1}(X) \in \mathbb{R}^d$. To quantify the approximation error, we define the following metric: for $1 \leqslant p \leqslant +\infty$,

$$\|\mathsf{H} - \mathsf{TF}\|_{L,p} := \left(\mathbb{E}_{X_L}[\|\mathsf{H}(X_L) - \mathsf{TF}_{-1}(X_L)\|_\infty^p]\right)^{1/p}. \tag{4}$$

The next theorem shows that a two-layer *single-head* transformer *without FFNs* suffices to implement vanilla induction heads.

**Theorem 3.1** (two-layer single-head $\mathsf{TF}$ w/o FFNs). *Let $\mathsf{IH}_2$ satisfy Eq. (3). Then exists an absolute constant $C > 0$ and a two-layer single-head transformer $\mathsf{TF}$ (without FFNs), with $D = 2d$, $W_K^{(1,1)} = W_Q^{(1,1)} = 0$, $p^{(2,1)} = 0$, and $\|W_K^{(2,1)}\|, \|W_Q^{(2,1)}\| \leqslant \mathcal{O}(1, \|W^\star\|_F)$, such that $\sup_{L\in\mathbb{N}^+} \|\mathsf{IH}_2 - \mathsf{TF}\|_{L,\infty} \leqslant \frac{C}{e^{p^{(1,1)}}}$.*

This theorem shows that single head suffices to approximate the vanilla induction head and moreover, the approximation efficiency is independent of the sequence length. The proof is provided in Appendix C.1, offering the following insights into how two-layer single-head transformers without FFNs implement vanilla induction heads:

- **The first layer** aggregates local tokens and outputs $z_s = [x_{s-1}, x_s]$ for the $s$-th token ($2 \leqslant s \leqslant L$). This is achieved by using $\mathsf{SA}$ with only RPE (no DP). Specifically, RPE allows $\mathsf{SA}$ to capture the *preceding token* via $x_{s-1} = \sum_{j\geqslant 1} x_{s-j}\rho(j)$ for each token $x_s$, where $\rho(\cdot) = \mathbb{I}\{\cdot = 1\}$. Hence, DP in this layer is not essential and can be omitted.

- **The second layer** extracts the relevant tokens using DP similarity. First, DP computes the similarity $\langle W_Q z_L, W_K z_s \rangle = x_L^\top W^\star x_{s-1}$, where $z_L = [x_{L-1}, x_L]$ and $z_s = [x_{s-1}, x_s]$ represent the hidden tokens output by the first layer. This similarity measure enables $\mathsf{SA}$ to identify tokens that match $x_L$. Subsequently, the value $W_V z_s$ extracts $x_s$, copying the subsequent token of $x_{s-1}$ and using it as the current prediction. In this layer, RPE is not necessary and can be omitted.

**Remark 3.2** (Alignment with experimental findings). Our theoretical analysis is consistent with the experimental observations reported in Elhage et al. (2021). Specifically, the experiments there demonstrate that $\mathsf{SA}$ in the first layer attends to adjacent tokens, while $\mathsf{SA}$ in the second layer retrieves information related to the current token. Our analysis identifies components responsible for these two operations, and reveals that *single-head* transformers suffice to perform them efficiently. Furthermore, we validate our theoretical construction through a *probing experiment*, presented in Figure 6.

## 3.2 Generalized Induction Heads: In-context $n$-gram and Generic Similarity

Although the standard induction head defined in Eq. (3) is intuitive, it exhibits notable limitations: **1)** it retrieves only a *single token*, potentially missing *complete local information* and leading to false retrievals; **2)** it relies solely on the *dot-product* to measure the similarity between two tokens, which is not sufficiently general.

**Formulation of $\mathsf{IH}_n$.** Motivated by the limitation **1)** above, we define a generalized induction head:

$$\mathsf{IH}_n(X_L) = \sum_{s=n}^{L-1} x_s \pi_s, \quad \pi_s = \mathrm{sm}\left(\left(X_{L-n+2:L}^\top W^\star X_{\nu-n+1:\nu-1}\right)_{\nu=n}^{L-1}\right)_{\nu=s}, \tag{5}$$

where the patch $X_{\nu-n+1:\nu-1}$ and $X_{L-n+2:L}$ incorporate *local information* near the previous token $x_{\nu-1}$ and the current $x_L$, respectively. This induction head operates based on the similarity between pairs: $(X_{s-n+1:s-1}; X_{L-n+2:L})$ for $s = n, \ldots, L-1$. Notably, $n$ is typically small when extracting *local semantics*, so we assume $n \leqslant 100$ in Eq. (5).[1]

Integrating richer local information facilitates more accurate information retrieval. The model (5) retrieves previous $(n-1)$-token patch that are similar to the current $(n-1)$-token patch, thereby generalizing the vanilla induction head (3), which considers only single-token retrieval ($n = 2$ in Eq. (5)). For example, as depicted in Figure 1, if the current local information is `The D` (comprising two tokens), and prior local information such as `Mr D` and `Mrs D` is identified as similar to `The D`, transformer would copy and paste their subsequent token, `urs`, as the prediction.

**Theorem 3.3** (two-layer multi-head TF w/o FFNs). *Let $\mathsf{IH}_n$ satisfy Eq. (5). Then, for any $H \in \mathbb{N}^+$ and rate $q \in \mathbb{N}^+$, there exists a constant $C_{n,q} > 0$ and a two-layer $H$-head transformer $\mathsf{TF}(\cdot)$ (without FFNs), with $D = nd$, such that:* $\sup_{L \in \mathbb{N}^+} \|\mathsf{IH}_n - \mathsf{TF}\|_{L,\infty} \leqslant \left(\frac{C_{n,q}}{H}\right)^q$, *where $C_{n,q} = \mathcal{O}(nq^2)$.*

This theorem demonstrates that two-layer multi-head transformers, even without FFNs, can *efficiently* implement the generalized induction head (5). Notably, the approximation error scales as $\mathcal{O}(H^{-q})$, where $q$ can be arbitrarily large. Moreover, for a fixed $q$, $H \geqslant \Omega(n)$ is sufficient to ensure a good approximation. The proof of this theorem is provided in Appendix C.2.

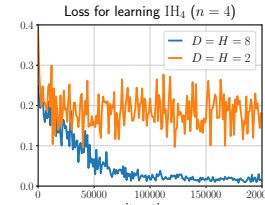

Figure 2: (Detailed in Fig. 7) The loss of training two-layer transformers with varying $H$ and $D$ to learn the target in Eq. (5) with $n = 4$. These indicate that the transformer with $H = D = 8$ ($> n$) successfully expresses this task, while the transformer with $H = D = 2$ ($< n$) fails.

**Experimental validation.** First, we validate our theoretical construction in Theorem 3.3 through a probing experiment, presented in Figure 6. Additionally, the **necessity** of the required number of heads $H$ and embedding dimension $D$ in Theorem 3.3 is experimentally verified in Figure 2. The detailed experimental results and setups are provided in Appendix B.

**The role of multiple heads.** In Theorem 3.3, multiple heads are employed in the first layer to approximate the $n$-gram interaction, represented by the $n-1$ memory kernels $\{\rho_j := \mathbb{I}\{\cdot = j\}\}_{j=1}^{n-1}$. Thus, TF can capture $n-1$ *preceding tokens* via $x_{s-j} = \sum_{k \geqslant 1} x_{s-k}\rho_j(k)$ for $j \in [n-1]$. Intuitively, as $n$ increases, more memory kernels are required for accurate approximation, necessitating more attention heads. In contrast, Theorem 3.1 only requires approximating a single memory kernel $\mathbb{I}\{\cdot = 1\}$, which can be efficiently achieved using a single attention head.

**Formulation of $\mathsf{GIH}_n$.** Building on the formulation (5), and motivated by the limitation **2)** above, we further consider the following generalized induction head:

$$\mathsf{GIH}_n(X_L) = \sum_{s=n}^{L-1} x_s \pi_s, \quad \pi_s = \mathrm{sm}\left(\left(\left(g\left(X_{L-n+2:L}; X_{\nu-n+1:\nu-1}\right)\right)_{\nu=n}^{L-1}\right)_{\nu=s}\right), \tag{6}$$

where $g : \mathbb{R}^{D \times (n-1)} \times \mathbb{R}^{D \times (n-1)} \to \mathbb{R}$ denotes a generic function measuring the similarity between two $(n-1)$-length patches.

This model retrieves previous relevant multi-token patch $X_{s-n+1:s-1}$ that is similar to the current multi-token patch $X_{L-n+2:L}$, utilizing the generalized similarity function $g(\cdot, \cdot)$. This mechanism is more general than Eq. (5), which is limited to dot-product similarities. For instance, the use of general similarity $g$ enables the model to recognize not only synonymous but also antonymic semantics, thereby improving both the accuracy and diversity of in-context retrievals.

**Theorem 3.4** (two-layer multi-head TF with FFNs). *Let $\mathsf{GIH}_n$ satisfy Eq. (6). Suppose the similarity function $g$ is $\alpha$-well-behaved (see Definition C.7). Then there exist two absolute constants $A, B > 0$ (only depending on the properties of $g$) such that: for any $H, M \in \mathbb{N}^+$ and rate $q \in \mathbb{N}^+$, there exists a constant $C_{n,q} > 0$ and a two-layer $H$-head transformer $\mathsf{TF}(\cdot)$ with FFNs of width $M$, such that*

$$\|\mathsf{GIH}_n - \mathsf{TF}\|_{L,2} \leqslant A\left(\frac{C_{n,q}}{H}\right)^q + B\frac{L^{1/(1+2\alpha)}}{M^{\alpha/(1+3\alpha)}}, \text{ where } C_{n,q} = \mathcal{O}(nq^2).$$

---

[1]For a given context $X_{1:L}$, Eq. (5) predicts by considering all past $n$-gram pairs $x_\nu | X_{\nu-n+1:\nu-1}$ (for $\nu \leqslant L-1$), making it a context-dependent statistic, and is therefore called *in-context $n$-gram*. Additionally, since $n$ used in classic $n$-grams is typical small (e.g., 2, 3, 4), so we make a reasonable assumption that $n \leqslant 100$.

This theorem establishes that if the similarity function $g$ is well-behaved, two-layer multi-head transformers with FFNs can efficiently implement the generalized induction head (6).

**The role of FFNs.** In contrast to Theorem 3.3, transformer models in Theorem 3.4 include FFNs. These FFN layers are used to approximate the similarity function $g$. Specifically, we consider the proper orthogonal decomposition (POD) of $g$, which can be viewed as an extension of the matrix singular value decomposition (SVD) applied to functions of two variables. For $g : \mathcal{I} \times \mathcal{I} \to \mathbb{R}$, its POD is $g(u,v) = \sum_{k=1}^{\infty} \sigma_k \phi_k(u) \psi_k(v)$, where $\phi_k, \psi_k$ are orthonormal bases for $L^2(\mathcal{I})$ (see Appendix E for details). Intuitively, the FFN in the first layer is used to efficiently approximate $K$ bases ($\phi_i$'s and $\psi_i$'s). Then, in the second layer, DP in SA can approximately reconstruct $g$ by using the truncated sum $g(u,v) \approx \sum_{k=1}^{K} \sigma_k \phi_k(u) \psi_k(v)$. The complete proof is deferred to Appendix C.3.

## 4 THE TRANSITION FROM LAZY TO RICH MECHANISMS IN LEARNING DYNAMICS

In this section, we investigate the dynamics of learning induction heads using a transformer, particularly focusing on how this differs from $n$-gram learning. To facilitate the analysis, we consider a mixed target function that comprises a 4-gram component and a vanilla induction head component as defined in Eq. (3). Specifically, we study the gradient flow dynamics of a two-layer multi-head transformer without FFNs on this task.

### 4.1 SETUPS

#### 4.1.1 MIXED TARGET FUNCTION

**Mixed target function.** Let the input sequence be $X = (x_1, \cdots, x_L) \in \mathbb{R}^{1 \times L}$. Our mixed target function $f^\star$ contains both a 4-gram component $f_{\mathsf{G}_4}^\star$ and an in-context 2-gram component $f_{\mathsf{IH}_2}^\star$:

$$f^\star(X) := \left( \frac{\alpha^\star}{1 + \alpha^\star} f_{\mathsf{G}_4}^\star(X), \frac{1}{1 + \alpha^\star} f_{\mathsf{IH}_2}^\star(X) \right)^\top \in \mathbb{R}^2, \tag{7}$$

where $\alpha^\star > 0$ represents the relative weight between the two components: $f_{\mathsf{G}_4}^\star(X)$ and $f_{\mathsf{IH}_2}^\star(X)$. Here, $f_{\mathsf{G}_4}^\star$ represents a 4-gram component and $f_{\mathsf{IH}_2}^\star$ is given by the vanilla induction head (3) to represent a type of in-context 2-gram information:

$$f_{\mathsf{G}_4}^\star(X) := x_{L-2}, \quad f_{\mathsf{IH}_2}^\star(X) := \sum_{s=2}^{L-1} x_s \, \mathrm{sm}\left( \left( x_L w^{\star 2} x_{\nu-1} \right)_{\nu=2}^{L-1} \right)_{\nu=s}.$$

Note that $f_{\mathsf{G}_4}^\star$ denotes a "simplest" 4-gram target, where the next token is predicted according to the conditional probability $p(z|X) = p(z|x_L, x_{L-1}, x_{L-2}) = \mathbb{I}\{z = x_{L-2}\}$.

**Remark 4.1** (The reason for considering 4-gram). Note that our target includes a 4-gram component rather than simpler 2- or 3-gram components. As suggested by the experimental results in Elhage et al. (2021), for a learned two-layer transformer that implements vanilla induction head $\mathsf{IH}_2$, the first layer has extracted both $x_L$ and $x_{L-1}$, which can be outputted using the residual block. Thus, the 2- and 3-gram targets: $p(z|X) = \mathbb{I}\{z = x_L\}$ and $p(z|X) = \mathbb{I}\{z = x_{L-1}\}$ must be learned prior to the induction head. Hence we focus on the more challenging 4-gram target to avoid trivializing the learning process, though our analysis extends straightforwardly to the 2- or 3-gram scenarios.

**Remark 4.2** (Extension). Since the transformer studied in this section does not have FFNs, its expressive power is limited. Consequently, we only consider the simple but representative mixed target (7). However, (7) can be generalized to $f^\star(X) = F(f_{\mathsf{G}_4}^\star(X); f_{\mathsf{IH}_2}^\star(X))$, where $F$ is general nonlinear function. Such a form can be efficiently approximated by transformers with FFNs. We leave the dynamics analysis under this general setting for future work.

#### 4.1.2 TWO-LAYER MULTI-HEAD TRANSFORMER WITH REPARAMETERIZATION

**Two-layer multi-head transformer w/o FFNs.** We consider a simple two-layer multi-head transformer TF, where the first layer contains a single head $\mathsf{SA}^{(1,1)}$, and the second layer contain two heads $\mathsf{SA}^{(2,1)}, \mathsf{SA}^{(2,2)}$. Given an input sequence $X = (x_1, \cdots, x_L) \in \mathbb{R}^{1 \times L}$, it is first embedded as $X^{(0)} := (X^\top, 0^\top) \in \mathbb{R}^{2 \times L}$. The model then processes the sequence as follows:

$$X^{(1)} = X^{(0)} + \mathsf{SA}^{(1,1)}(X^{(0)}), \quad \mathsf{TF}(X) = \mathsf{SA}^{(2,1)}(X^{(1)}) + \mathsf{SA}^{(2,2)}(X^{(1)}).$$

**Reparameterization.** Despite the simplification, the transformer above is still too complicated for dynamics analysis. To overcome this challenge, we adopt the reparametrization trick used in previous works (Tian et al., 2023; Huang et al., 2023; Chen et al., 2024b). Specifically, by Theorem 3.1 and its proof, *the first layer does not require DP, and the second layer does not require RPE*. Moreover, to express the 4-gram component $f^\star_{\mathsf{G}_4}$, we only need an additional head without DP in the second layer. Therefore, we can reparameterize the model as follows, (see Appendix D.1 for details):

- **First layer.** This layer consists of a single attention head without DP. The only trainable parameter is $p^{(1,1)}$, which governs the RPE component.

- **Second layer.** This layer contains two heads and five trainable parameters. The first head without DP is responsible to fit $f^\star_{\mathsf{G}_4}$ using parameters $p^{(2,1)}, w^{(2,1)}_V$, while the second head without RPE is responsible to fit $f^\star_{\mathsf{IH}_2}$ with parameters $w^{(2,2)}_V, w^{(2,2)}_K, w^{(2,2)}_Q$.

The set of all six trainable parameters across both layers is denoted by $\theta$.

### 4.1.3 Gradient Flow on Square Loss

We consider the Gaussian input and square loss, both of which are commonly used in analyzing transformer dynamics and ICL (Akyürek et al., 2022; Huang et al., 2023; Wang et al., 2024). The loss is defined as:

$$\mathcal{L}(\theta) = \frac{1}{2}\mathbb{E}_{X \sim \mathcal{N}(0, I_{L \times L})}\Big[\|\mathsf{TF}_{-1}(X; \theta) - f^\star(X)\|_2^2\Big], \tag{8}$$

To characterize the learning of $\mathsf{G}_4$ and $\mathsf{IH}_2$, we introduce the following two partial losses:

$$\mathcal{L}_{\mathsf{G}_4}(\theta) = \frac{1}{2}\mathbb{E}_X\left(\mathsf{TF}_{-1,1}(X;\theta) - f^\star_1(X)\right)^2, \ \mathcal{L}_{\mathsf{IH}_2}(\theta) = \frac{1}{2}\mathbb{E}_X\left(\mathsf{TF}_{-1,2}(X;\theta) - f^\star_2(X)\right)^2,$$

which correspond to the two dimensions in $\mathsf{TF}_{-1}(X; \theta) - f^\star(X) \in \mathbb{R}^2$, respectively. It follows that $\mathcal{L}(\theta) = \mathcal{L}_{\mathsf{G}_4}(\theta) + \mathcal{L}_{\mathsf{IH}_2}(\theta)$.

**Gradient flow (GF).** We analyze the GF for minimizing the objective (8):

$$\frac{\mathrm{d}\theta(t)}{\mathrm{d}t} = -\nabla\mathcal{L}(\theta(t)), \ \ \text{starting with } \theta(0) = (\sigma_{\mathrm{init}}, \cdots, \sigma_{\mathrm{init}})^\top, \tag{9}$$

where $0 < \sigma_{\mathrm{init}} \ll 1$ is sufficiently small. Note that $\sigma_{\mathrm{init}} \neq 0$ prevents $\nabla\mathcal{L}(\theta(0)) = 0$.

**Layerwise training paradigm**. We consider a layerwise training paradigm in which, during each stage, only one layer is trained by GF. Specifically,

- **Training Stage I:** In this phase, only the parameter in the first layer, i.e., $p^{(1,1)}$, is trained.

- **Training Stage II:** In this phase, the first layer parameter $p^{(1,1)}$ keeps fixed and only parameters in the second layer are trained: $w^{(2,1)}_V, w^{(2,2)}_V, p^{(2,1)}, w^{(2,2)}_Q, w^{(2,2)}_K$.

This type of layerwise training has been widely used to study the training dynamics of neural networks, including FFN networks (Safran and Lee, 2022; Bietti et al., 2023; Wang et al., 2023) and transformers (Tian et al., 2023; Nichani et al., 2024; Chen et al., 2024b).

**Lemma 4.3** (Training Stage I). *For the Training Stage I,* $\lim\limits_{t \to +\infty} p^{(1,1)}(t) = +\infty$.

According to (16), this lemma implies that, at the end of Training Stage I, the first layer captures the preceding token $x_{s-1}$ for each token $x_s$, i.e., $y_s = x_{s-1}$. This property is crucial for transformers to implement induction heads and aligns with our approximation result in Theorem 3.1. The proof of Lemma 4.3 is deferred to Appendix D.2.

### 4.2 Training Stage II: Transition from 4-gram to Induction Head

In this section, we analyze the dynamics in Training Stage II. We start from the following lemma:

**Lemma 4.4** (Parameter balance). *In Training Stage II, it holds that* $|w^{(2,2)}_Q(t)|^2 \equiv |w^{(2,2)}_K(t)|^2$.

Lemma 4.4 is similar to the balance result for homogeneous networks (Du et al., 2018), and its proof can be found at the start of Appendix D.3. By this lemma, we can define $w_{KQ}^{(2,2)} := w_Q \equiv w_K$. Additionally, Lemma 4.3 ensures that $p^{(1,1)} = +\infty$ holds during Stage II. For simplicity, we denote $w_{V_1} := w_V^{(2,1)}, w_{V_2} := w_V^{(2,2)}, p := p^{(2,1)}, w_{KQ} := w_{KQ}^{(2,2)}$. Consequently, the training dynamics are reduced to four parameters

$$\theta = (w_{V_1}, w_{V_2}, p, w_{KQ}),$$

where we still denote the set of parameters as $\theta$ without introducing ambiguity. It is important to note that the problem remains **highly non-convex** due to the joint optimization of both inner parameters $(p, w_{KQ})$ and outer parameters $(w_{V_1}, w_{V_2})$ in the two heads. At this training stage, GF has a **unique fixed point** $w_{V_1} = \frac{\alpha^\star}{1+\alpha^\star}, w_{V_2} = \frac{1}{1+\alpha^\star}, p = +\infty, w_{KQ} = w^\star$, which corresponds to a global minimizer of the objective (8).

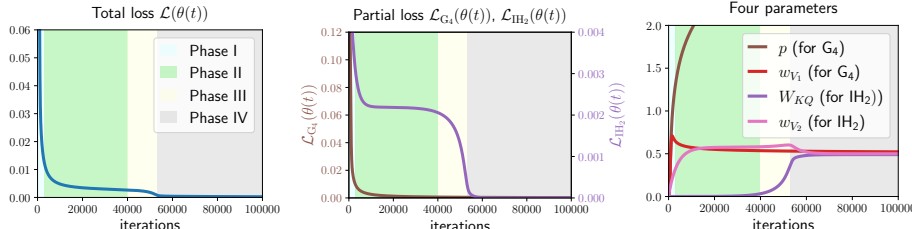

Figure 3: Visualization of the dynamical behavior of Training Stage II with total loss, partial loss, and the parameter evolution. Here, $\alpha^\star = 1, w^\star = 0.49, \sigma_{\text{init}} = 0.01, L = 40$. The is clearly shown that transformer learns the 4-gram component first and then, starts to learn the induction head mechanism. Notably, the entire dynamics unfold in four distinct phases, consistent with our theoretical results (Theorem 4.5). For more experimental details, we refer to Appendix B.3.

As shown in Figure 3, a learning transition from the 4-gram mechanism to the induction head mechanism does occur in our setting. Moreover, the learning process exhibits a four-phase dynamics. The next theorem provides a precise characterization of the four phases, whose proof can be found in Appendix D.3.

**Theorem 4.5** (Learning transition and 4-phase dynamics). *Let $\alpha^\star = \Omega(1)$ and $w^\star = \mathcal{O}(1)$, and we consider the regime of small initialization ($0 < \sigma_{\text{init}} \ll 1$) and long input sequences ($L \gg 1$). Then we have the following results:*

- **Phase I (partial learning).** *In this phase, most of the 4-gram component in the mixed target is learned, while a considerable number of induction head component have not yet been learned. Specifically, let $T_{\text{I}} = \mathcal{O}(1)$, then we have the following estimates:*

$$\mathcal{L}_{\mathsf{G}_4}(\theta(T_{\text{I}})) \leqslant 0.01 \cdot \mathcal{L}_{\mathsf{G}_4}(\theta(0)), \ \mathcal{L}_{\mathsf{IH}_2}(\theta(T_{\text{I}})) \geqslant 0.99 \cdot \mathcal{L}_{\mathsf{IH}_2}(\theta(0)).$$

- **Phase II (plateau) + Phase III (emergence).** *In these two phases, the learning of the induction head first gets stuck in a plateau for $T_{\text{II}}$ time, then is learned suddenly. Specifically, denoted by an observation time $T_o = \Theta(L)$, we have the following tight estimate of the duration:*

$$T_{\text{II}} := \inf \left\{ t > T_o : \mathcal{L}_{\mathsf{IH}_2}(\theta(t)) \leqslant 0.99 \cdot \mathcal{L}_{\mathsf{IH}_2}(\theta(T_o)) \right\} = \Theta\left( (\alpha^\star + 1)^2 L \log(1/\sigma_{\text{init}})/w^{\star 2} \right);$$

$$T_{\text{III}} := \inf \left\{ t > T_o : \mathcal{L}_{\mathsf{IH}_2}(\theta(t)) \leqslant 0.01 \cdot \mathcal{L}_{\mathsf{IH}_2}(\theta(T_o)) \right\} = \Theta\left( (\alpha^\star + 1)^2 L \log(1/\sigma_{\text{init}})/w^{\star 2} \right).$$

*During these phases, the parameter $w_{KQ}$ (for learning $w^\star$ in $\mathsf{IH}_2$) increases exponentially:*

$$w_{KQ}(t) = \sigma_{\text{init}} \exp\left( \Theta\left( \frac{w^{\star 2} t}{(1 + \alpha^\star)^2 L} \right) \right), \ t < T_{\text{III}}.$$

- **Phase IV (convergence).** *In this phase, the loss converges toward zero. Specifically, the following convergence rates hold for all $t > T_{\text{III}}$:*

$$\mathcal{L}_{\mathsf{G}_4}(\theta(t)) = \mathcal{O}\left( \frac{1}{t} \right), \ \mathcal{L}_{\mathsf{IH}_2}(\theta(t)) = \mathcal{O}\left( \exp\left( -\Omega\left( \frac{w^{\star 2} t}{(1 + \alpha^\star)^2 L} \right) \right) \right),$$

*and $\mathcal{L}(\theta(t)) = \mathcal{L}_{\mathsf{G}_4}(\theta(t)) + \mathcal{L}_{\mathsf{IH}_2}(\theta(t))$.*

By this theorem, the 4-gram mechanism is first learned, taking time $T_{\mathrm{I}}$. Then, the learning of the induction head mechanism enters a plateau, taking time $T_{\mathrm{II}}$, followed by a sudden emergence of learning, taking time $T_{\mathrm{III}} - T_{\mathrm{II}}$. Finally, the loss for both components converges to zero.

**The clear learning transition.** When any one of $L, \alpha^\star, 1/\sigma_{\mathrm{init}}, 1/w^\star$ is sufficiently large, Phase II lasts for $T_{\mathrm{II}} \gg 1$. During this phase, the 4-gram component has been learned well but the induction head component remains underdeveloped, demonstrating a distinct learning transition. Moreover, Theorem 4.5 and its proof reveal two key factors that drive this transition:

- **Time-scale separation due to high- and low-order parameter dependence in self attention**. The learning of DP and RPE components differ in their parameter dependencies. DP component exhibits a quadratic dependence on the parameter $w_{KQ}$, while RPE component shows linear dependence on the parameter $p$. With small initialization $\sigma_{\mathrm{init}} \ll 1$, a clear time-scale separation emerges: $|\dot{w}_{KQ}| \sim w_{KQ} \ll 1$ (DP, slow dynamics) and $|\dot{p}| \sim 1$ (RPE, fast dynamics). Consequently, the induction head (fitted by DP) is learned much slower than the 4-gram component (fitted by RPE). This time-scale separation accounts for the term $\log(1/\epsilon_{\mathrm{init}})$ in the plateau time $T_{\mathrm{II}}$.

- **Speed difference due to component proportions in the mixed target.** The 4-gram target component and the induction-head component have differing proportions in the mixed target. A simple calculation shows: $\mathcal{L}_{\mathsf{G}_4}(0) \sim \alpha^{\star 2}/(1+\alpha^\star)^2$; If $w^\star = \mathcal{O}(1)$, then $\mathcal{L}_{\mathsf{IH}_2}(0) \sim 1/[(1+\alpha^\star)^2 L]$. Notably, $\mathcal{L}_{\mathsf{IH}_2}(0)$ is significantly smaller than $\mathcal{L}_{\mathsf{G}_4}(0)$. This proportion disparity accounts for the $(1+\alpha^\star)^2 L$ term in the plateau time $T_{\mathrm{II}}$.

**Proof idea.** We highlight that our fine-grained analysis of entire learning process is guided by two key observations: 1) the dynamics of the two heads can be decoupled; 2) there exist a distinct transition point in the dynamics of each head, as shown in Figure 3 (right). These insights lead us to divide the analysis of each head into two phases: a monotonic phase and a convergence phase. Particularly, for the convergence phase, we introduce a novel Lyapunov function that leverages the unique dynamical structure of self-attention. This Lyapunov function may be of independent interest and offers potential for studying broader issues in self-attention dynamics.

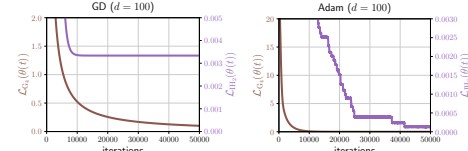

Figure 4: (Detailed in Fig. 10) Partial loss for the experiment comparing **GD v.s. Adam optimizer** in high-dimensional settings ($D = 100$). It is shown that Adam experiences a *challenging transition characterized by *multiple plateaus* during learning induction heads. This finding closely resembles the dynamics for GD.

**Experimental validation.** We conduct additional experiments to validate our theoretical insights into the training dynamics and learning transition across a wider range of scenarios. These includes using *discrete data distribution* (Figure 9), *other optimization algorithms* (Figure 4) in high-dimensional settings, as well as training *real-world* transformers on natural language datasets (Figure 5). The detailed experimental results and setups are provided in Appendix B

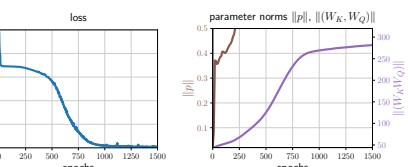

Figure 5: (Detailed in Fig. 8) The loss and parameter norm for the experiment training a two-layer two-head **standard transformer** (without any simplification) on the **wikitext-2** dataset (Merity et al., 2016). This further supports our theoretical insights regarding the loss plateau, as well as the time-scale separation between the learning of RPE and the DP structure.

## 5 CONCLUSION

In this work, we present a comprehensive theoretical analysis of how transformers implement induction heads, examining both the approximation and optimization aspects. From the approximation standpoint, we identify the distinct roles of each transformer component in implementing induction heads of varying complexity. On the optimization side, we analyze a toy setting, where we clearly characterize how learning transitions from $n$-grams to induction heads. Looking forward, an important direction for future research is to investigate the dynamics of learning general induction heads, which are crucial for realizing stronger ICL capabilities.

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

# Appendix

## A  RELATED WORKS

**Empirical observations of induction head.** The induction head mechanism was first identified by Elhage et al. (2021) in studying how two-layer transformers perform language modeling. Subsequently, Olsson et al. (2022) conducted a more systematic investigation, revealing two key findings: 1) induction head emerges abruptly during training, and 2) induction head plays a critical role in the development of in-context learning capabilities. To obtain a fine-grained understanding of how induction head emerges during training, recent studies have developed several synthetic settings (Reddy, 2024; Edelman et al., 2024; Bietti et al., 2024). Particularly, Bietti et al. (2024) successfully reproduced the fast learning of (global) bigrams and the slower development of induction head. Despite these efforts, a comprehensive theoretical understanding of how the induction head operates in two-layer transformers and how it is learned during training remains elusive.

**Expressiveness of transformers.** Theoretically, Dehghani et al. (2019); Pérez et al. (2021); Wei et al. (2022) explored the Turing-completeness of transformers; Yun et al. (2019) established the universal approximation property of transformers. Subsequent studies examined the efficiency of transformers in representing specific functions or tasks, such as sparse functions (Edelman et al., 2022), targets with nonlinear temporal kernels (Jiang and Li, 2023), practical computer programs (Giannou et al., 2023), long but sparse memories (Wang et al., 2024), induction head (Sanford et al., 2024a;b; Rajaraman et al., 2024), and memorization and reasoning (Chen and Zou, 2024). Besides, many studies

suggest that transformers achieve in-context learning by approximating gradient-based iterations across various layers (Garg et al., 2022; Akyürek et al., 2022; Von Oswald et al., 2023; Mahankali et al., 2023; Bai et al., 2023; Shen et al., 2023). Besides, several studies explored the limitation of transformer's expressivity, particularly in modeling formal languages or simulating circuits (Hahn, 2020; Weiss et al., 2021; Bhattamishra et al., 2020; Merrill et al., 2022; Merrill and Sabharwal, 2023). Among all these works, the most closely related to ours are Rajaraman et al. (2024), which examined a generalized induction head similar to our Eq. (5). Specifically, they showed that multi-layer transformers with single-head attention can implement this mechanism. In contrast, we prove that two-layer transformers are sufficient if multihead attention is used.

**Training dynamics of transformers.** To gain insights into the dynamics of training transformers, several studies have analyzed simplified transformers on toy tasks. These tasks include learning distinct/common tokens (Tian et al., 2023), leaning balance/inblanced features (Huang et al., 2023), linear regression task (Zhang et al., 2023; Ahn et al., 2024), multi-task linear regression (Chen et al., 2024a), binary classification (Li et al., 2024), transformer with diagonal weights (Abbe et al., 2024), learning causal structure (Nichani et al., 2024), sparse token selection task (Wang et al., 2024), and learning $n$-gram Markov chain (Chen et al., 2024b). Additionally, studies such as those by Ataee Tarzanagh et al. (2023), Tarzanagh et al. (2023) and Vasudeva et al. (2024) have analyzed scenarios where transformers converge to max-margin solutions. Furthermore, Thrampoulidis (2024) has examined the implicit bias of next-token prediction. Among these works, the most closely related to ours are Nichani et al. (2024) and Chen et al. (2024b), which proved that two-layer transformers can converge to induction head solutions. In this work, we explore a setting where the target is a mixture of 4-gram and induction head. We show that two-layer transformers can effectively converge to this mixed target and provide a precise description of the learning process associated with each component. Importantly, we are able to capture the *abrupt transition* from learning 4-gram patterns to mastering the induction head mechanism—a critical phase in the learning of induction heads, as highlighted in the seminal works (Elhage et al., 2021; Olsson et al., 2022).

Now we discuss the relationship between our work and two closely related studies (Bietti et al., 2024; Edelman et al., 2024).

**Comparison with Bietti et al. (2024).**

- **Approximation analysis:**

  Bietti et al. (2024) focus primarily on the implementation of the vanilla induction head. In contrast, our study extends this analysis by investigating not only how two-layer transformers achieve vanilla induction heads (Eq. (3)) but also how they implement generalized induction heads, i.e., in-context n-grams (Eqs. (5) and (6)).

  Furthermore, our work provides explicit approximation rate results, offering insights into the distinct roles of multiple heads, positional encoding, dot-product structure, and FFNs in implementing these induction heads.

- **Optimization analysis:**

  *Study objective:* While Bietti et al. (2024) examines the transition from 2-gram to induction head, our work focuses on the transition from 4-gram to induction head.

  *study methods:* Bietti et al. (2024) conducts extensive experiments supported by partial theoretical properties but does not fully characterize the training dynamics theoretically. In contrast, our study provides **a precise theoretical analysis of the entire training process** in a toy model, uncovering the sharp transition from 4-gram to induction head.

  *Main insights:* Bietti et al. (2024) emphasizes the the role of weight matrices as associative memories and the impact of data distributional properties. Our analysis, on the other hand, identifies two primary drivers of the transition: (1) the time-scale separation due to low- and high-order parameter dependencies in self-attention; (2) the speed differences caused by the relative proportions of the two components in the mixed target.

**Comparison with Edelman et al. (2024).** The primary connection between Edelman et al. (2024) and our work lies in the optimization analysis. Specifically, Edelman et al. (2024) focuses on the transition from uni-gram to bi-gram mechanisms in Markov Chain data. In contrast, our study investigates the transition from 4-gram to in-context 2-gram mechanisms (induction head). Additionally, we theoretically identify two primary drivers of the transition: (1) the time-scale separation due to low-

and high-order parameter dependencies in self-attention; (2) the speed differences caused by the relative proportions of the two components in the mixed target.

**Comparison with Rajaraman et al. (2024).** Recently, Theorem 4.4 in Rajaraman et al. (2024) explored a generalized induction head similar to Eq. (5) and showed that multi-layer transformers, with an artificial positional encoding $\mathbb{I}\{\cdot = k\}, (k \in [n])$, can implement it. In contrast, our Theorem 3.3 demonstrates that two layers suffice if multi-head self-attention with **Alibi** positional encoding is adopted. Estimating the number of attention heads needed is a key contribution of our result.

# B  EXPERIMENTS

## B.1  EXPERIMENTS SUPPORTING APPROXIMATION RESULTS

1. Supporting our **construction** in Theorem 3.3.

**Setup.** We linear probing experiments (Alain and Bengio, 2016) on the transformers with $H = D = 8$ trained in the above experiment (Figure 7). For each checkpoint model TF, we denote its output in the first layer on the input sequence $X$ as $\text{TF}^{(1)}(X)$. The probing loss is measured by

$$\text{dist}\left(X_{\cdot -n+1:\cdot}; \text{TF}^{(1)}(X)\right) = \min_{P \in \mathbb{R}^{D \times n}} : \sum_{s=n}^{L} \left\| X_{s-n+1:s} - \text{TF}_s^{(1)}(X)P \right\|, \text{ where } n = 4, L = 10,$$

and $X = (x_1, \cdots, x_L)$ is generated by $x_i \overset{i.i.d.}{\sim} \text{Unif}(\{\pm 1\})$ with testing batch 1000. The results are shown in Figure 6.

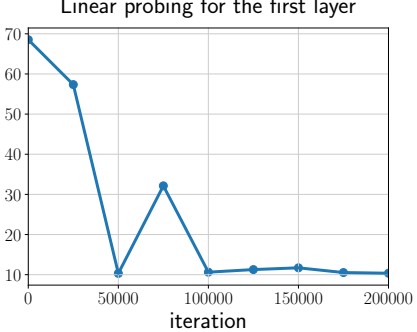

Figure 6: Probing results supporting our **construction** in Theorem 3.3. First, we train a two-layer two-layer transformer with head $H = 8$ and embedding dimension $D = 8$ to learn Eq. (5) with $n = 4$, and the checkpoints are stored during training. For each checkpoint model TF, we denote its output in the *first layer* on the input sequence $X$ as $\text{TF}^{(1)}(X)$. To validate whether it encodes the semantic information $X_{s-n+2:s}$ near each $x_s$, as predicted by our construction, we conduct a standard linear probing experiment (Alain and Bengio, 2016). Specifically, we measured

$$\text{dist}\left(X_{\cdot -n+1:\cdot}; \text{TF}^{(1)}(X)\right) = \min_{P \in \mathbb{R}^{D \times n}} : \sum_{s=n}^{L} \left\| X_{s-n+1:s} - \text{TF}_s^{(1)}(X)P \right\|. \text{ As the results shown, the}$$

probing loss decreases significantly during training, confirming our key construction in Theorem 4.3: **the first layer is responsible for extracting local semantic information $X_{s-n+2:s}$ near each $x_s$,** enabling the second layer to generate the final output.

2. Supporting the **necessity** of the required $H$ and $D$ in Theorem 3.3.

**Setup.** We train two-layer transformers (without FFN layers) with varying $H$ and $D$ to learn the generalized induction head (5) with $n = 4$. The input sequence $X = (x_1, \cdots, x_L)$ is boolean, with $x_i \overset{i.i.d.}{\sim} \text{Unif}(\{\pm 1\})$ and $L = 10$. Each model is trained for 200,000 iterations using squared loss and (online) Adam optimizer with learning rate $5\text{e}{-}4$ and batch size $B = 100$. Both layers are trained simultaneously. The results for the models with $D = H = 8$ and $D = H = 2$ are presented in Figure 8.

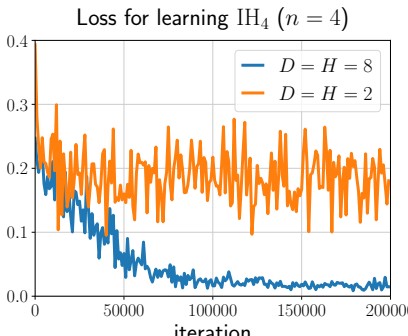

Figure 7: Results supporting the **necessity** of the required number of heads $H$ and embedding dimension $D$ in Theorem 3.3. We train two-layer transformers with varying $H$ and $D$ to learn the target in Eq. (5) with $n = 4$. The results indicate that the transformer with $H = D = 8\ (> n)$ successfully expresses this task, while the transformer with $H = D = 2\ (< n)$ fails. These results confirm that the sufficient conditions provided in Theorem 3.3 ($H \gtrsim n$ and $D \geqslant nd$, where $d = 1$ in our setting) are also nearly necessary.

### B.2 ADDITIONAL EXPERIMENTS SUPPORTING OPTIMIZATION DYNAMICS

### 1. Standard transformers on real-world natural language dataset.

**Setup.** We train a two-layer two-head **standard transformer** with Alibi RPE (without any simplification) on the **wikitext-2** dataset, a natural language dataset (Merity et al., 2016). The transformer has an embedding dimension $D = 128$ and FFN width $W = 512$. For this dataset, the input dimension is $d = 33278$. We use a context length $L = 200$ and batch size $B = 32$. The parameters are initialized with the scale 0.01. The model is trained for 1,500 epochs on 1 H100, using cross-entropy loss and SGD with learning rate 0.1, and the initialization scale is 0.01. It is important to note that **both layers are trained simultaneously**. The results are presented in Figure 8.

### 2. Discrete token distribution in toy setting.

**Setup.** We modified the Gaussian input distribution used in the setup for Figure 3 to a boolean input distribution, where each input token, where each input token $x_i \overset{iid}{\sim} \text{Unif}(\{\pm 1\})$ for $i \in [L]$, All other experimental setups remain the same as in the setup for Figure 3. The training dynamics of Stage (ii) are presented in Figure 9. We can see clearly that the dynamical behavior of the learning process is nearly the same as the one observed for Gaussian inputs in Figure 3.

### 3. Adam in high-dimensional toy setting.

**Setup.** We modified the setup for Figure 3 to employ a high-dimensional model ($D = 100$). Specifically, the target is $w^\star = 0.49 I_D/D$, the dot-produce parameters are $W_K, W_Q \in \mathbb{R}^D$, initialized such that $\|W_K\|_F, \|W_Q\|_F = \sigma_{\text{init}}$. Additionally, for the Adam optimizer, we use learning rate $5\text{e}{-}4$. All other experimental setups remain the same as in the setup for Figure 3.

The training dynamics are depicted in Figure 10, where, for comparison, results using GD are also presented. In both scenarios, the learning process begins with the 4-gram pattern, followed by a gradual learning phase of the induction head mechanism. Notably, within the given number of iterations, GD remains stuck in the plateau, whereas Adam successfully escapes that plateau.

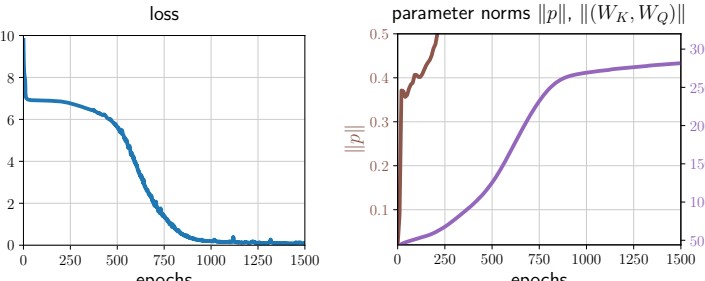

Figure 8: The loss and parameters for the experiment training a two-layer two-head **standard transformer** (without any simplification) on the **wikitext-2** dataset (Merity et al., 2016). Here, $\|p\|$ and $\|(W_K, W_Q)\|$ denote the Frobenius norms of all positional encoding parameters and all $W_K, W_Q$ parameters across layers and heads, respectively, The results show that: the loss exhibits a clear plateau; position encoding $p$'s are learned first; and the dot-product structure $W_K, W_Q$ are learned slowly at the beginning, resembling an exponential increase; additionally, as $W_K, W_Q$ are learned, the loss escapes that plateau. These findings closely resemble the behavior observed in our toy model (Figure 3). This experiment provides further support for our theoretical insights regarding the **time-scale separation** between the learning of positional encoding and the dot-product structure.

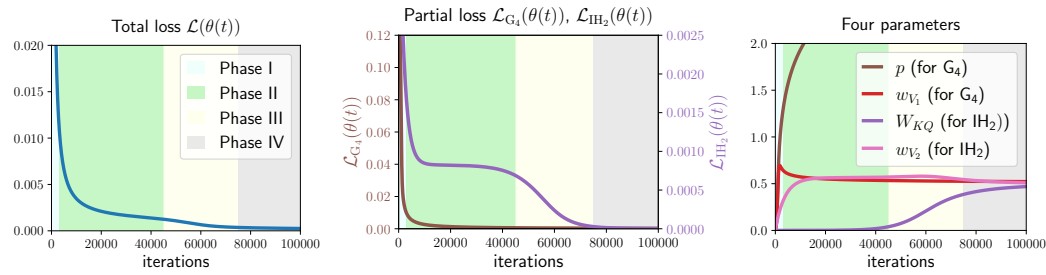

Figure 9: Visualization of the total loss, partial loss, and the parameter dynamics, for the experiment on **discrete token distribution** (Boolean, $X \sim \mathrm{Unif}(\{\pm 1\}^L)$) in our toy setting with $\alpha^\star = 1, w^\star = 0.49, \sigma_{\text{init}} = 0.01, L = 40$. The figure clearly shows that transformer learns the 4-gram component first and then, starts to learn the induction head mechanism. Notably, the entire dynamics exhibit four phases. These results are **extremely similar** to that observed with Gaussian inputs, as shown in Figure 3.

### B.3 EXPERIMENTAL DETAILS FOR FIGURE 3

In line with our theoretical setting, we examine a simplified two-layer transformer, as described in Alibi RPE. Specifically, the first layer only contains RPE and the second layer consists of two heads: one uses only RPE and the other employs only dot-product structure. The target function is specified by (7) with $\alpha^\star = 1, w^\star = 0.49, \sigma_{\text{init}} = 0.01, L = 40$, and the distribution of each token is Gaussian, i.e., $x_i \overset{iid}{\sim} \mathcal{N}(0, 1)$ for $i \in [L]$. Training is conducted by minimizing the squared loss (8) using online SGD with learning rate 0.1 and batch size $B = 1,000$. Following our theoretical analysis, the two layers are trained sequentially:

- Training Stage I: only the first layer is trained for 100,000 iterations;

- Training Stage II: Subsequently, only the second layer undergoes training for another 100,000 iterations.

The dynamical behavior of the Training Stage II is visualized in Figure 3.

**Compute resources.** Real-world experiments on wikitext-2 are conducted on 1 A100 GPU, while other synthetic experiments are conducted on CPU.

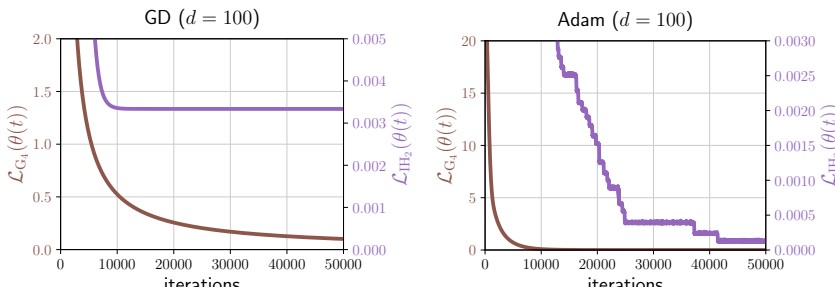

Figure 10: Partial loss for the experiment comparing **GD v.s. Adam optimizer** in high-dimensional settings ($D = 100$). In this setting, a larger $D$ increases the difficulty of the transition from the lazy regime (learning 4-gram) to the rich regime (learning induction head). The results indicate that: (1) GD learns the 4-gram component first but becomes stuck in a plateau when learning induction head; (2) Adam, while eventually transitioning from the lazy regime (learning 4-gram) to the rich regime (learning induction head), experiences a **challenging** transition characterized by **multiple plateaus** during learning induction heads. This finding closely resembles the dynamics for GD.

## C  PROOFS IN SECTION 3

### C.1  PROOF OF THEOREM 3.1

$$\mathsf{IH}_2(X_L) = (x_s)_{s=2}^{L-1} \operatorname{softmax}\left(\left(x_L^\top W^\star x_{s-1}\right)_{s=2}^{L-1}\right)^\top, \tag{10}$$

**Theorem C.1** (Restatement of Theorem 3.1). *Let $\mathsf{IH}_2$ satisfy Eq. (10). Then, there exists a constant $C > 0$ and a two-layer single-head transformer* TF *(without FFNs), with $D = 2d$, $W_K^{(1,1)} = W_Q^{(1,1)} = 0$, $p^{(2,1)} = 0$, and $\|W_K^{(2,1)}\|, \|W_Q^{(2,1)}\| \leqslant \mathcal{O}(1, \|W^\star\|_F)$, such that*

$$\sup_{L \in \mathbb{N}^+} \|\!\|\mathsf{IH}_2 - \mathsf{TF}\|\!\|_{L,\infty} \leqslant \frac{C}{e^{p^{(1,1)}}}.$$

*Proof.* We consider two-layer single-head transformer without FFN, where the first layer has the residual block, while the second layer does not have the residual block.

We first embed each token into $\mathbb{R}^D$ as $\begin{pmatrix} x_s \\ 0 \end{pmatrix}$ and take $W_V^{(1)} = \begin{pmatrix} 0 & 0 \\ I_{d\times d} & 0 \end{pmatrix}$, then the $s$-th output token of the first layer is

$$\begin{pmatrix} x_s \\ y_s \end{pmatrix} = \begin{pmatrix} x_s \\ (x_\tau)_{\tau=1}^{s-1} \operatorname{softmax}\left(\left(-p^{(1,1)}(s-1-\tau)\right)_{\tau=1}^{s-1}\right)^\top \end{pmatrix}.$$

Then for the second layer, we choose $p^{(2,1)} = 0$,

$$W_Q^{(2,1)} = \begin{pmatrix} 0 & 0 \\ I_{d\times d} & 0 \end{pmatrix}, \ W_K^{(2,1)} = \begin{pmatrix} 0 & 0 \\ 0 & W^\star \end{pmatrix}, \ W_V^{(2,1)} = \begin{pmatrix} I_{d\times d} & 0 \\ 0 & 0 \end{pmatrix} \in \mathbb{R}^{D\times D},$$

and the projection $W_O^{(2)} = (I_{d\times d} \ \ 0_{d\times d}) \in \mathbb{R}^{d\times D}$.

Then the last output token of the second layer is

$$(x_s)_{s=2}^{L-1} \operatorname{softmax}\left(\left(x_L^\top W^\star y_s\right)_{s=2}^{L-1}\right)^\top.$$

By Lemma E.1 , for any $L \in \mathbb{N}^+$

$$\|\!\|\mathsf{IH}_2 - \mathsf{TF}\|\!\|_{L,\infty}$$
$$= \sup_{X_L} \|\mathsf{IH}(X_L) - \mathsf{TF}_{-1}(X_L)\|_\infty$$

$$= \left\| (x_s)_{s=2}^{L-1} \, \mathrm{softmax}\left( \left( x_L^\top W^\star y_s \right)_{s=2}^{L-1} \right)^\top - (x_s)_{s=2}^{L-1} \, \mathrm{softmax}\left( \left( x_L^\top W^\star x_{s-1} \right)_{s=2}^{L-1} \right)^\top \right\|_\infty$$

$$\leqslant \| (x_s)_{s=2}^{L-1} \|_{\infty,\infty} \left\| \mathrm{softmax}\left( \left( x_L^\top W^\star y_s \right)_{s=2}^{L-1} \right)^\top - \mathrm{softmax}\left( \left( x_L^\top W^\star x_{s-1} \right)_{s=2}^{L-1} \right)^\top \right\|_1$$

$$\leqslant 2 \sup_{2 \leqslant s \leqslant L-1} \left| x_L^\top W^\star y_s - x_L^\top W^\star x_{s-1} \right|$$

$$\leqslant 2 \| x_L^\top W^\star \|_1 \sup_s \| y_s - x_{s-1} \|_\infty$$

$$\leqslant 2 \sum_{i,j} |W_{i,j}^\star| \sup_s \left\| (x_\tau)_{\tau=1}^{s-1} \, \mathrm{softmax}\left( \left( -p^{(1,1)}(s-1-\tau) \right)_{\tau=1}^{s-1} \right)^\top - x_{s-1} \right\|_\infty$$

$$\leqslant 2 \| W^\star \|_{1,1} \sup_s \left\| \mathrm{softmax}\left( \left( -p^{(1,1)}(s-1-\tau) \right)_{\tau=1}^{s-1} \right)^\top - \boldsymbol{e}_{s-1} \right\|_1$$

$$= 2 \| W^\star \|_{1,1} \sup_{2 \leqslant s \leqslant L-1} \frac{\sum_{\tau=1}^{s-2} \exp\left( -p^{(1,1)}(s-1-\tau) \right)}{\sum_{\tau=1}^{s-1} \exp\left( -p^{(1,1)}(s-1-\tau) \right)}$$

$$\leqslant 2 \| W^\star \|_{1,1} \lim_{s \to +\infty} \frac{\sum_{\tau=1}^{s-2} \exp\left( -p^{(1,1)}(s-1-\tau) \right)}{\sum_{\tau=1}^{s-1} \exp\left( -p^{(1,1)}(s-1-\tau) \right)} = 2 \| W^\star \|_{1,1} e^{-p^{(1,1)}}.$$

$$\square$$

### C.2 Proof of Theorem 3.3

$$\mathsf{IH}_n(X_L) = (x_s)_{s=n}^{L-1} \, \mathrm{softmax}\left( \left( X_{L-n+2:L}^\top W^\star X_{s-n+1:s-1} \right)_{s=n}^{L-1} \right)^\top, \tag{11}$$

**Theorem C.2** (Restatement of Theorem 3.3). *Let* $\mathsf{IH}_n$ *satisfy Eq. (11). Then for any* $H \in \mathbb{N}^+$ *and rate* $q \in \mathbb{N}^+$, *there exists a constant* $C_{n,q} > 0$ *and a two-layer* $H$*-head transformer* $\mathsf{TF}(\cdot)$ *(without FFNs), with* $D = nd$, *such that:*

$$\sup_{L \in \mathbb{N}^+} \| \mathsf{IH}_n - \mathsf{TF} \|_{L,\infty} \leqslant \left( \frac{C_{n,q}}{H} \right)^q,$$

*where* $C_{n,q} = \mathcal{O}(nq^2)$.

*Proof.* We consider two-layer multi-head transformer without FFN, where the first layer has the residual block, while the second layer does not have the residual block.

First, we choose the embedding dimension $D = nd$, and parameters in the embedding map

$$W_E = \begin{pmatrix} I_{d \times d} \\ 0_{(D-d) \times d} \end{pmatrix} \in \mathbb{R}^{D \times d}, \quad b_E = 0 \in \mathbb{R}^D,$$

then each token $x_s^{(0)}$ after embedding is

$$x_s^{(0)} = W_E x_s + b_E = \begin{pmatrix} x_s \\ 0 \end{pmatrix} \in \mathbb{R}^D.$$

This proof can be summarized as the following process for $\mathsf{TF}_{-1}$:

$$(x_s)_{s=n}^{L-1} \, \mathrm{softmax}\left( \left( \hat{X}_{L-n+2:L}^\top W^\star \hat{X}_{s-n+1:s-1} \right)_{s=n}^{L-1} \right)^\top$$

Step II. 2-st Attn $\uparrow$

$$(x_L^\top, \hat{x}_{L-1}, \ldots, \hat{x}_{L-n+1})^\top$$

Step I. 1-st Attn $\uparrow$

$$(x_L^\top, 0^\top)^\top$$

Additionally, in this proof, the following projection matrices are used:

$$P_i := \left(0_{d \times (i-1)d}, I_{d \times d}, 0_{d \times (D-id)}\right) \in \mathbb{R}^{d \times D}, \quad i \in [n].$$

**Step I. The first layer.**    We use 1-st Attn with residual to copy the previous tokens $(x_{s-n+1}, \cdots, x_{s-1})$ of each token $x_s$. We use $H = \sum_{i=1}^{n-1} H_i$ attention heads to realize this step.

By lemma E.2, for any rate $q \in \mathbb{N}^+$, there exists a function

$$\phi_i^{\exp}(t) = \sum_{h=1}^{H_i} \alpha_{h,i} e^{-\beta_{h,i}(t-1)}$$

such that $\beta_h > 0$ and

$$\|\mathbb{I}\{\cdot = i\} - \phi_i^{\exp}(\cdot)\|_{\ell_1(\mathbb{N})} = \sum_{s=i}^{+\infty} |\mathbb{I}\{s = 1\} - \phi^{\exp}(s)| \leqslant C \frac{A^q (q^2)^q e^{0.01(q+1)i}}{H_i^q},$$

where $A, C > 0$ are absolute constants.

For $h = \sum_{j=1}^{i-1} H_j, 1 + \sum_{j=1}^{i-1} H_j, \ldots, \sum_{j=1}^{i} H_j$, we choose parameters as follows

$$p^{(1,h)} = \beta_{h,i}, \quad W_V^{(1,h)} = \alpha_{h,i} \left(\sum_{j=0}^{H_i} \exp(-\beta_{h,i}(j-1))\right) S_i,$$

$$W_K^{(1,h)} = W_Q^{(1,h)} = 0, \quad W_O^{(1)} = I_{D \times D}$$

where $S_i \in \mathbb{R}^{D \times D}$ is a shift matrix that takes out the first $d$ elements of a vector and shifts it backward to the $(id+1)$-th to $(i+1)d$-th elements. Then

$$\left(P_{i+1} \sum_{h=\sum_{j=1}^{i-1} H_j}^{\sum_{j=1}^{i} H_j} \mathsf{SA}^{(1,h)}(X_L^{(0)})\right)_{-1} = \sum_{h=\sum_{j=1}^{i-1} H_j}^{\sum_{j=1}^{i} H_j} \alpha_{h,i} \sum_{s=1}^{L-1} e^{-\beta_{h,i}(s-1)} x_{L-s}.$$

We denote the $s$-th output token of the first layer

$$x_s^{(1)} := \mathsf{SA}^{(1)} \left(X_{0:s}^{(0)}\right)_{-1},$$

Then the approximation error of this step is

$$\varepsilon_{\mathsf{SA}}^{(1)} := \sup_s \left\|x_s^{(1)} - X_{s-n+1:s}\right\|_\infty \leqslant \sup_s \left(\|x_s - x_s\|_\infty + \sum_{i=1}^{n-1} \left\|P_{i+1} x_s^{(1)} - x_{s-i}\right\|_\infty\right)$$

$$= \sup_s \sum_{i=1}^{n-1} \left\|P_{i+1} x_s^{(1)} - x_{s-i}\right\|_\infty \leqslant \sup_s \sum_{i=1}^{n-1} \|\mathbb{I}\{\cdot = i\} - \phi_i^{\exp}(\cdot)\|_{\ell_1(\mathbb{N})}$$

$$\leqslant C(Aq^2)^q \sum_{i=1}^{n-1} \frac{e^{0.01(q+1)i}}{H_i^q}.$$

Consequently, one detail is to assign the head number $\{H_i\}_{i=1}^n$ such that the error's sum $\sum_{i=1}^{n-1} \frac{e^{0.01(q+1)i}}{H_i^q}$ is as small as possible. Our way is solving the minimization problem

$$\min : \sum_{i=1}^{n-1} \frac{e^{0.01(q+1)i}}{H_i^q}$$

$$\text{s.t.} \sum_{i=1}^{n-1} H_i = H,$$

which suggests that we should choose the head number:

$$H_i = \frac{e^{0.01i}}{\sum_{j=1}^{n-1} e^{0.01j}}, i \in [n-1].$$

Thus, we obtain the bound

$$\varepsilon_{\mathsf{SA}}^{(1)} \leqslant \frac{C(Aq^2)^q}{H^q} \left( \sum_{i=1}^{n-1} e^{0.01i} \right)^q \leqslant C \left( \frac{Aq^2 n e^{0.01n}}{H} \right)^q.$$

Noticing $n < 100$, we have:

$$\varepsilon_{\mathsf{SA}}^{(1)} \leqslant C \left( \frac{eAq^2 n}{H} \right)^q.$$

We focus on **the case of large $H$**:

$$H \geqslant CAenq^2,$$

which ensures $\varepsilon_{\mathsf{SA}}^{(1)} \leqslant 1$. Therefore, $x_s^{(1)} \in [-2, 2]^D$:

$$\left\| x_s^{(1)} \right\|_\infty \leqslant \sup_s \left\| x_s^{(1)} - X_{s-n+1:s} \right\|_\infty + \| X_{s-n+1:s} \|_\infty \leqslant \varepsilon_{\mathsf{SA}}^{(1)} + 1 \leqslant 2, \quad \forall s.$$

**Step II. The second layer.** For the second Attn, we only need use the first head (by setting $W_V^{(2,h)} = 0$ for $h \geqslant 1$). Specifically, we choose $p^{(2,1)} = 0$,

$$W_Q^{(2,1)} = \begin{pmatrix} 0 & 0 \\ I_{(D-d)\times(D-d)} & 0 \end{pmatrix}, \ W_K^{(2,1)} = \begin{pmatrix} 0 & 0 \\ 0 & W^\star \end{pmatrix}, \ W_V^{(2)} = \begin{pmatrix} I_{d\times d} & 0 \\ 0 & 0 \end{pmatrix} \in \mathbb{R}^{D\times D},$$

and the projection $W_O^{(2)} = \begin{pmatrix} I_{d\times d} & 0_{(D-d)\times d} \end{pmatrix} \in \mathbb{R}^{d\times D}$.

For simplicity, we use the following notations:

$$\hat{X}_{s-n+1:s-1} := W_Q = \begin{pmatrix} P_2 x_s^{(1)} \\ \vdots \\ P_n x_s^{(1)} \end{pmatrix}, \ \text{ for } n \leqslant s \leqslant L-1; \qquad \hat{X}_{L-n+2:L} := \begin{pmatrix} P_1 x_L^{(1)} \\ \vdots \\ P_{n-1} x_L^{(1)}. \end{pmatrix}.$$

Then the output of the second layer is

$$x_L^{(2)} = (x_s)_{s=n}^{L-1} \operatorname{softmax}\left( \left( \hat{X}_{L-n+2:L}^\top W^\star \hat{X}_{s-n+1:s-1} \right)_{s=n}^{L-1} \right)^\top$$

By using these bounds and Lemma E.1,

$$\left\| x_L^{(2)} - (x_s)_{s=n}^{L-1} \operatorname{softmax}\left( \left( X_{L-n+2:L}^\top W^\star X_{s-n+1:s-1} \right)_{s=n}^{L-1} \right)^\top \right\|_\infty$$

$$\leqslant \sum_{s=n}^{L-1} \left| \operatorname{softmax}\left( \left( \hat{X}_{L-n+2:L}^\top W^\star \hat{X}_{s-n+1:s-1} \right)_{s=n}^{L-1} \right)^\top - \operatorname{softmax}\left( \left( X_{L-n+2:L}^\top W^\star X_{s-n+1:s-1} \right)_{s=n}^{L-1} \right)^\top \right) \right|$$

$$\leqslant 2 \max_s \left| \hat{X}_{L-n+2:L}^\top W^\star \hat{X}_{s-n+1:s-1} - X_{L-n+2:L}^\top W^\star X_{s-n+1:s-1} \right|$$

$$\leqslant 2 \left( \max_s \left| \left( \hat{X}_{L-n+2:L} - X_{L-n+2:L} \right)^\top W^\star \hat{X}_{s-n+1:s-1} \right| + \max_s \left| X_{L-n+2:L}^\top W^\star \left( \hat{X}_{s-n+1:s-1} - X_{s-n+1:s-1} \right) \right| \right)$$

$$\leqslant 2 \left( 2\|W^\star\|_{(1,1)} \left\| \hat{X}_{L-n+2:L} - X_{L-n+2:L} \right\|_\infty + \|W^\star\|_{1,1} \max_s \left\| \hat{X}_{s-n+1:s-1} - X_{s-n+1:s-1} \right\|_\infty \right)$$

$$\leqslant 6\|W^\star\|_{1,1} \cdot \varepsilon_{\mathsf{SA}}^{(1)}.$$

Since the above inequality holds for any $L$ and $X_L$, we obtain our bound:

$$\sup_{L\in\mathbb{N}^+} \|\mathsf{IH}_n - \mathsf{TF}\|_{L,\infty} \leqslant 6C\|W^\star\|_{(1,1)} \left( \frac{6eAnq^2}{H} \right)^q \leqslant \left( \frac{6C\|W^\star\|_{1,1} eAnq^2}{H} \right)^q,$$

For **the remained case**, $H < CAenq^2$, we can simply choose TF with all $0$ parameters. Then the approximation error can be trivially bounded by:

$$\sup_{L \in \mathbb{N}^+} \|\|\mathsf{IH}_n - \mathsf{TF}\|\|_{L,\infty} = \sup_{L \in \mathbb{N}^+} \|\|\mathsf{IH}_n - 0\|\|_{L,\infty} \leqslant 1.$$

Then, the two cases can be unified by:

$$\sup_{L \in \mathbb{N}^+} \|\|\mathsf{IH}_n - \mathsf{TF}\|\|_{L,\infty} \leqslant \begin{cases} 6C\|W^\star\|_{1,1} \left(\frac{eAnq^2}{H}\right)^q, \ H \geqslant CAenq^2 \\ 1, \ \text{otherwise} \end{cases} \leqslant \left(\frac{(6\|W^\star\|_{1,1} + 1)\, CeAnq^2}{H}\right)^q.$$

In the theorem, we can choose $C_{n,q} = (6\|W^\star\|_{1,1} + 1)\, CeAnq^2$, which satisfies $C_{n,q} \lesssim nq^2$. $\qquad \square$

## C.3 PROOF OF THEOREM 3.4

### C.3.1 APPROXIMATION RESULTS FOR FFNS

Since the setting in this subsection includes FFNs, we introduce the following preliminary results about the approximation of FFNs.

The well-known universal approximation result for two-layer FNNs asserts that two-layer FNNs can approximate any continuous function (Barron, 1992; 1993; 1994). Nonetheless, this result lacks a characterization of the approximation efficiency, i.e., how many neurons are needed to achieve a certain approximation accuracy? Extensive pre-existing studies aimed to address this gap by establishing approximation rates for two-layer FFNs. A representative result is the Barron theory (E et al., 2019; 2021; Ma et al., 2020): any function $f$ in Barron space $\mathcal{B}$ can be approximated by a two-layer FFN with $M$ hidden neurons can approximate $f$ efficiently, at a rate of $\mathcal{O}(\|f\|_{\mathcal{B}}/\sqrt{M})$. This rate is remarkably independent of the input dimension, thus avoiding the Curse of Dimensionality. Specifically, Barron space is defined in as follows:

**Definition C.3** (Barron space (E et al., 2019; 2021; Ma et al., 2020))**.** Consider functions $f : X \to \mathbb{R}$ that admit the following representation: $f(x) = \int_\Omega a\sigma(b^\top x + c)\rho(\mathrm{d}a, \mathrm{d}b, \mathrm{d}c)$, $x \in X$. For any $p \in [1, +\infty]$, we define the Barron norm as $\|f\|_{\mathcal{B}_p} := \inf_\rho \left(\mathbb{E}_\rho\left[|a|^p(\|b\|_1 + |c|)^p\right]\right)^{1/p}$. Then the Barron space are defined as: $\mathcal{B}_p := \{f \in \mathcal{C} : \|f\|_{\mathcal{B}_p} < +\infty\}$.

**Proposition C.4** (E et al. (2019))**.** *For any* $p \in [1, +\infty]$, $\mathcal{B}_p = \mathcal{B}_\infty$ *and* $\|f\|_{\mathcal{B}_p} = \|f\|_{\mathcal{B}_\infty}$.

**Remark C.5.** From the Proposition above, the Barron spaces $\mathcal{B}_p$ are equivalent for any $p \in [1, +\infty]$. Consequently, in this paper, we use $\mathcal{B}$ and $\|\cdot\|_{\mathcal{B}}$ to denote the Barron space and Barron norm.

The next lemma illustrates the approximation rate of two-layer FFNs for Barron functions.

**Lemma C.6** (Ma et al. (2020))**.** *For any* $f \in \mathcal{B}$, *there exists a two-layer ReLU neural network* $\mathsf{FFN}(x) = \sum_{k=1}^{M} a_w \sigma(b_k^\top x + c_k)$ *with* $M$ *neurons such that*

$$\|f - \mathsf{FFN}\|_{L^\infty[-2,2]} \leqslant \mathcal{O}\left(\frac{\|f\|_{\mathcal{B}} \sqrt{\log M}}{\sqrt{M}}\right).$$

### C.3.2 PROPER ORTHOGONAL DECOMPOSITION

Proper orthogonal decomposition (POD) can be viewed as an extension of the matrix singular value decomposition (SVD) applied to functions of two variables. Specifically, for a square integrable function $g : \mathcal{I} \times \mathcal{I} \to \mathbb{R}$, it has the following decomposition (Theorem 3.4 in Yarvin and Rokhlin (1998), Theorem VI.17 in Reed and Simon (1980)):

$$g(u, v) = \sum_{k=1}^{\infty} \sigma_k \phi_k(u) \psi_k(v). \tag{12}$$

Here, $\phi_k, \psi_k$ are orthonormal bases for $L^2(\mathcal{I})$, and $\sigma_k \geqslant 0$ are the singular values, arranged in descending order.

Recently, Jiang and Li (2023) also used POD to study the approximation rate of single-layer single-head Transformer for the targets with nonlinear temporal kernels.

Given that two-layer FFNs can efficiently approximate Barron functions (Ma et al., 2020), which is dense in $L^2([0,1]^d)$ (Siegel and Xu, 2020), we introduce the following technical definition regarding the well-behavior POD, which is used for our theoretical analysis.

**Definition C.7** (Well-behaved POD). Let the POD of $g : [-2,2]^D \times [-2,2]^D \mapsto \mathbb{R}$ be $g(u,v) = \sum_{k=1}^{\infty} \sigma_k \phi_k(u)\psi_k(v)$. We call the function $g$ has $\alpha$-well-behaved POD ($\alpha > 0$) if:

- The decay rate of singular values satisfies $\sigma_k = \mathcal{O}(1/k^{1+\alpha})$;

- The $L^\infty$ norms, Barron norms, and Lipschitz norms of the POD bases are all uniformly bounded:
$\sup_k \left( \|\phi_k\|_{L^\infty} \vee \|\psi_k\|_{L^\infty} \vee \|\phi_k\|_{\mathcal{B}} \vee \|\psi_k\|_{\mathcal{B}} \vee \|\phi_k\|_{\mathrm{Lip}} \vee \|\psi_k\|_{\mathrm{Lip}} \right) < \infty.$

### C.3.3 PROOF OF THEOREM 3.4

$$\mathsf{GIH}_n(X_L) = (x_s)_{s=n}^{L-1} \, \mathrm{softmax}\Big( \big( g\big(X_{L-n+2:L}; X_{s-n+1:s-1}\big)\big)_{s=n}^{L-1} \Big)^\top, \tag{13}$$

**Theorem C.8** (Restatement of Theorem 3.4). *Suppose the similarity function $g$ is $\alpha$-well-behaved (see Definition C.7). Then there exist two absolute constants $A, B > 0$ (only depending on the properties of $g$) such that: for any $H, M \in \mathbb{N}^+$ and rate $q \in \mathbb{N}^+$, there exists a constant $C_{n,q} > 0$ and a two-layer $H$-head transformer $\mathsf{TF}(\cdot)$ with FFNs of width $M$, such that*

$$\|\mathsf{GIH}_n - \mathsf{TF}\|_{L,2} \leqslant A \left( \frac{C_{n,q}}{H} \right)^q + B \frac{L^{1/(1+2\alpha)}}{M^{\alpha/(1+3\alpha)}},$$

*where $C_{n,q} = \mathcal{O}(nq^2)$.*

*Proof.* We consider two-layer multi-head transformer with FFN, where the first layer has the residual block.

First, we set an constant $K \in \mathbb{N}^+$, and we will optimize it finally. We choose the embedding dimension $D = nd + 2(n-1)K$.

Additionally, in this proof, the following projection matrices are used:

$$P_i := \big( 0_{d \times (i-1)d}, I_{d \times d}, 0_{d \times (D-id)} \big) \in \mathbb{R}^{d \times D}, \quad i \in [n].$$

The proof sketch can be summarized as follows:

$$(x_s)_{s=n}^{L-1} \, \mathrm{softmax}\Big( \big( g\big(X_{L-n+2:L}; X_{s-n+1:s-1}\big)\big)_{s=n}^{L-1} \Big)^\top$$

$$\text{Step III. 2-st Attn } \uparrow$$

$$\Big( x_L^\top, \dots, \hat{x}_{L-n+1}, \hat{\phi}_1(\hat{X}_{L-n+2:L}), \dots, \hat{\phi}_K(\hat{X}_{L-n+2:L}), \hat{\psi}_1(\hat{X}_{L-n+1:L-1}), \dots, \hat{\psi}_K(\hat{X}_{L-n+1:L-1}) \Big)^\top$$

$$\text{Step II. 1-st FFN } \uparrow$$

$$(x_L^\top, \hat{x}_{L-1}, \dots, \hat{x}_{L-n+1}, 0^\top)^\top$$

$$\text{Step I. 1-st Attn } \uparrow$$

$$(x_L^\top, 0^\top)^\top$$

Recalling Definition C.7, there exists constants $C_g^\infty, C_g^{\mathcal{B}}, C_g^{\mathrm{Lip}} > 0$ such that:

$$\sup_k \left( \|\phi_k\|_\infty \vee \|\psi_k\|_\infty \right) \leqslant C_g^\infty, \ \sup_k \left( \|\phi_k\|_{\mathcal{B}} \vee \|\psi_k\|_{\mathcal{B}} \right) \leqslant C_g^{\mathcal{B}}, \ \sup_k \left( \|\phi_k\|_{\mathrm{Lip}} \vee \|\psi_k\|_{\mathrm{Lip}} \right) \leqslant C_g^{\mathrm{Lip}}.$$

Additionally, $\sigma_k = \mathcal{O}(1/k^{1+\alpha})$ implies that there exits a $C_\alpha > 0$ such that:

$$\sum_{k=K}^{\infty} \sigma_k < \frac{C_\alpha}{K^\alpha}, \quad \forall K \geqslant 1.$$

**Step I: Error in 1-st Attn layer.** This step is essentially the same as Step I in the proof of Theorem 3.3, so we write down the estimate of the first Attn layer directly: there exists absolute constants $C_1, C_2 > 0$ such that

$$\varepsilon_{\mathsf{SA}}^{(1)} := \sup_s \left\| x_s^{(1)} - X_{s-n+1:s} \right\|_\infty \leqslant C_2 \left( \frac{C_1 n q^2}{H} \right)^q.$$

Similar to the proof of Theorem 3.3, we first focus on **the case of large $H$:**

$$H \geqslant C_1 C_2 e n q^2,$$

which ensures $\varepsilon_{\mathsf{SA}}^{(1)} \leqslant 1$. Therefore, $x_s^{(1)} \in [-2, 2]^D$:

For simplicity, we use the following notations:

$$\hat{X}_{s-n+1:s-1} := W_Q = \begin{pmatrix} P_2 x_s^{(1)} \\ \vdots \\ P_n x_s^{(1)} \end{pmatrix}, \ \text{ for } n \leqslant s \leqslant L-1; \qquad \hat{X}_{L-n+2:L} := \begin{pmatrix} P_1 x_L^{(1)} \\ \vdots \\ P_{n-1} x_L^{(1)}. \end{pmatrix}.$$

**Step II: Error in 1-st FFN layer.** The 1-st FFN is used to approximate $\phi_k, \psi_k$ $(k = 1, \ldots, K)$. Each function is approximated by a 2-layer neural networks with $\frac{M}{2K}$ neurons defined on $\mathbb{R}^D$, and the FFNs are concatenated together ( refer to section 7.1 "Parallelization" in Schmidt-Hieber et al. (2020) ) as $\mathsf{FFN}^{(1)}$. We denote them as

$$\hat{\phi}_k(y) = \sum_{m=1}^{\frac{M}{2K}} a_m^k \sigma(b_m^{k\top} y + c_m^k)$$

$$\hat{\psi}_k(y) = \sum_{m=1}^{\frac{M}{2K}} \tilde{a}_m^k \sigma(\tilde{b}_m^{k\top} y + \tilde{c}_m^k)$$

Then by lemma C.6, such FFNs exist and satisfy the following properties hold for all $1 \leqslant k \leqslant K$:

$$\|\hat{\phi}_k - \phi_k\|_{L^\infty([-2,2]^D)} \leqslant \mathcal{O}\left( \|\phi_k\|_{\mathcal{B}} \sqrt{\frac{K \log M}{M}} \right) \leqslant \epsilon_{\mathsf{FFN}}^{(1)},$$

$$\|\hat{\psi}_k - \psi_k\|_{L^\infty([-2,2]^D)} \leqslant \mathcal{O}\left( \|\psi_k\|_{\mathcal{B}} \sqrt{\frac{K \log M}{M}} \right) \leqslant \epsilon_{\mathsf{FFN}}^{(1)},$$

where

$$\epsilon_{\mathsf{FFN}}^{(1)} := \mathcal{O}\left( C_g^{\mathcal{B}} \sqrt{\frac{K \log M}{M}} \right), \quad C_g^{\mathcal{B}} = \max\{\|\phi_k\|_{\mathcal{B}}, \|\psi_k\|_{\mathcal{B}}\}.$$

**Step III: Error in 2nd Attn layer.**

We use matrices in the second layer to take out elements needed

$$W_V^{(2)} = (I_{d \times d}, 0_{d \times D}) \in \mathbb{R}^{d \times D},$$

$$W_K^{(2,1)} = \sum_{i=k}^{K} \sqrt{\sigma_k} e_{k,(n-1)d+k} \in \mathbb{R}^{D \times D},$$

$$W_Q^{(2,1)} = \sum_{k=1}^{K} \sqrt{\sigma_k} e_{k,(n-1)d+K+k} \in \mathbb{R}^{D \times D}.$$

We denote the rank-$K$ truncation of $g$ as

$$g_K := \sum_{k=1}^{K} \sigma_k \phi_k \psi_k,$$

and its approximation as

$$\hat{g}_K := \sum_{k=1}^{K} \sigma_k \hat{\phi}_k \hat{\psi}_k$$

The second FFN is set to be identity map and we denote the final output as

$$x_L^{(2)} := (x_s)_{s=n}^{L-1} \operatorname{softmax}\Big(\big(\hat{g}_K\big(\hat{X}_{L-n+2:L}; \hat{X}_{s-n+1:s-1}\big)\big)_{s=n}^{L-1}\Big)^{\top},$$

First, we consider the error under the first norm, $\|\cdot\|_\infty$, which can be divided the total error into three components:

$$\left\| x_L^{(2)} - (x_s)_{s=n}^{L-1} \operatorname{softmax}\Big(\big(g\big(X_{L-n+2:L}; X_{s-n+1:s-1}\big)\big)_{s=n}^{L-1}\Big)^{\top} \right\|_\infty$$

$$\leqslant \left\| \operatorname{softmax}\Big(\big(\hat{g}_K\big(\hat{X}_{L-n+2:L}; \hat{X}_{s-n+1:s-1}\big)\big)_{s=n}^{L-1}\Big)^{\top} - \operatorname{softmax}\Big(\big(g\big(X_{L-n+2:L}; X_{s-n+1:s-1}\big)\big)_{s=n}^{L-1}\Big)^{\top} \right\|_\infty$$

$$\leqslant \left\| \operatorname{softmax}\Big(\big(\hat{g}_K\big(\hat{X}_{L-n+2:L}; \hat{X}_{s-n+1:s-1}\big)\big)_{s=n}^{L-1}\Big)^{\top} - \operatorname{softmax}\Big(\big(g_K\big(\hat{X}_{L-n+2:L}; \hat{X}_{s-n+1:s-1}\big)\big)_{s=n}^{L-1}\Big)^{\top} \right\|_\infty$$

$$+ \left\| \operatorname{softmax}\Big(\big(g_K\big(\hat{X}_{L-n+2:L}; \hat{X}_{s-n+1:s-1}\big)\big)_{s=n}^{L-1}\Big)^{\top} - \operatorname{softmax}\Big(\big(g_K\big(X_{L-n+2:L}; X_{s-n+1:s-1}\big)\big)_{s=n}^{L-1}\Big)^{\top} \right\|_\infty$$

$$+ \left\| \operatorname{softmax}\Big(\big(g_K\big(X_{L-n+2:L}; X_{s-n+1:s-1}\big)\big)_{s=n}^{L-1}\Big)^{\top} - \operatorname{softmax}\Big(\big(g\big(X_{L-n+2:L}; X_{s-n+1:s-1}\big)\big)_{s=n}^{L-1}\Big)^{\top} \right\|_\infty$$

$$\leqslant \max_s \left| \hat{g}_K\big(\hat{X}_{L-n+2:L}, \hat{X}_{s-n+1:s-1}\big) - g_K\big(\hat{X}_{L-n+2:L}, \hat{X}_{s-n+1:s-1}\big) \right|$$

$$+ \max_s \left| g_K\big(\hat{X}_{L-n+2:L}, \hat{X}_{s-n+1:s-1}\big) - g_K\big(X_{L-n+2:L}, X_{s-n+1:s-1}\big) \right|$$

$$+ \sum_{s=n}^{L-1} \left| g_K\big(X_{L-n+2:L}, X_{s-n+1:s-1}\big) - g\big(X_{L-n+2:L}, X_{s-n+1:s-1}\big) \right|$$

$$\tag{14}$$

For the first term in RHS of (14), it holds that:

$$\max_s \left| \hat{g}_K\big(\hat{X}_{L-n+2:L}, \hat{X}_{s-n+1:s-1}\big) - g_K\big(\hat{X}_{L-n+2:L}, \hat{X}_{s-n+1:s-1}\big) \right|$$

$$\leqslant \max_s \sum_{k=1}^{K} \sigma_k \left| \hat{\phi}_k(\hat{X}_{L-n+2:L}) \hat{\psi}_k(\hat{X}_{s-n+1:s-1}) - \phi_k(\hat{X}_{L-n+2:L}) \psi_k(\hat{X}_{s-n+1:s-1}) \right|$$

$$\leqslant \sum_{k=1}^{K} \sigma_k \left( \left\| \hat{\phi}_k \right\|_{L^\infty} \left\| \hat{\psi}_k - \psi_k \right\|_{L^\infty} + \|\psi_k\|_{L^\infty} \left\| \hat{\phi}_k - \phi_k \right\|_{L^\infty} \right)$$

$$\leqslant \epsilon_{\mathsf{FFN}}^{(1)} \cdot \sum_{k=1}^{K} \sigma_k \left( \left\| \hat{\phi}_k \right\|_{L^\infty} + \|\psi_k\|_{L^\infty} \right)$$

$$\leqslant \epsilon_{\mathsf{FFN}}^{(1)} \cdot \sum_{k=1}^{K} \sigma_k \left( \|\phi_k\|_{L^\infty} + \left\| \hat{\phi}_k - \phi_k \right\|_{L^\infty} + \|\psi_k\|_{L^\infty} \right)$$

$$\leqslant \epsilon_{\mathsf{FFN}}^{(1)} \cdot (2C_g^\infty + 1) \sum_{k=1}^{K} \sigma_k \leqslant (2C_g^\infty + 1) C_\alpha \epsilon_{\mathsf{FFN}}^{(1)}.$$

For the second term in RHS of (14), we have:

$$\max_s \left| g_K\big(\hat{X}_{L-n+2:L}, \hat{X}_{s-n+1:s-1}\big) - g_K\big(X_{L-n+2:L}, X_{s-n+1:s-1}\big) \right|$$

$$\leqslant \max_s \sum_{k=1}^{K} \sigma_k \Big( \|\phi_k\|_{L^\infty} |\psi_k(\hat{X}_{s-n+1:s-1}) - \hat{\psi}_k(X_{s-n+1:s-1})|$$

$$+ \|\psi_k\|_{L^\infty} |\phi_k(\hat{X}_{L-n+1:L-1}) - \phi_k(X_{L-n+1:L-1})|\Big)$$

$$\leqslant \max_s \sum_{k=1}^K \sigma_k \Big( \|\phi_k\|_{L^\infty} \|\psi_k\|_{\mathrm{Lip}} \Big\| \hat{X}_{s-n+1:s-1} - X_{s-n+1:s-1} \Big\|$$

$$+ \|\psi_k\|_{L^\infty} \|\phi_k\|_{\mathrm{Lip}} \Big\| \hat{X}_{L-n+1:L-1} - X_{L-n+1:L-1} \Big\| \Big)$$

$$\leqslant 2C_g^\infty C_g^{\mathrm{Lip}} \epsilon_{\mathsf{SA}}^{(1)} \cdot \left( \max_s \sum_{k=1}^K \sigma_k \right) \leqslant 2C_g^\infty C_g^{\mathrm{Lip}} C_\alpha \epsilon_{\mathsf{SA}}^{(1)}.$$

Additioanlly, the third term in RHS of (14), its $L^2$ holds that:

$$\int_{[0,1]^{d\times L}} \left( \sum_{s=n}^{L-1} \left| g_K\big(X_{L-n+2:L}, X_{s-n+1:s-1}\big) - g\big(X_{L-n+2:L}, X_{s-n+1:s-1}\big) \right| \right)^2 dX$$

$$\leqslant (t-1-n) \sum_{s=n}^{L-1} \int_{[0,1]^{D\times L}} \left| g_K\big(X_{-n+2:t}, X_{s-n+1:s-1}\big) - g\big(X_{L-n+2:L}, X_{s-n+1:s-1}\big) \right|^2 dX$$

$$= (L-1-n)^2 \int_{[0,1]^D \times [0,1]^D} |g(u,v) - g_K(u;v)|^2 \, \mathrm{d}u\mathrm{d}v$$

$$= (L-1-n)^2 \int \left( \sum_{k=K+1}^{+\infty} \sigma_k \phi_k(u)\psi_k(v) \right)^2 du \, dv$$

$$\leqslant \int \left( \sum_{k=K+1}^{+\infty} \sigma_k \phi_k^2(u) \right) \left( \sum_{k=K+1}^{+\infty} \sigma_k \psi_k^2(v) \right) du \, dv$$

$$\leqslant (L-1-n)^2 \left( \sum_{k=K+1}^{\infty} \sigma_k \right)^2 \leqslant \frac{(L-1-n)^2 C_\alpha^2}{K^{2\alpha}}.$$

Now we combine three error terms together to obtain the total $L^2$ error for the output of this layer:

$$\int_{X\in[0,1]^{d\times L}} \left\| x_L^{(2)} - (x_s)_{s=n}^{L-1} \operatorname{softmax}\Big( \big(g(X_{L-n+2:L}; X_{s-n+1:s-1})\big)_{s=n}^{L-1}{}^\top \Big) \right\|_\infty^2 \mathrm{d}X$$

$$\leqslant 3 \int_{X\in[0,1]^{d\times L}} \max_s \left| \hat{g}_K\big(\hat{X}_{L-n+2:L}, \hat{X}_{s-n+1:s-1}\big) - g_K\big(\hat{X}_{L-n+2:L}, \hat{X}_{s-n+1:s-1}\big) \right|^2 \mathrm{d}X$$

$$+ 3 \int_{X\in[0,1]^{d\times L}} \max_s \left| g_K\big(\hat{X}_{L-n+2:L}, \hat{X}_{s-n+1:s-1}\big) - g_K\big(X_{L-n+2:L}, X_{s-n+1:s-1}\big) \right|^2 \mathrm{d}X$$

$$+ 3 \int_{X\in[0,1]^{d\times L}} \left( \sum_{s=n}^{L-1} \left| g_K\big(X_{L-n+2:L}, X_{s-n+1:s-1}\big) - g\big(X_{L-n+2:L}, X_{s-n+1:s-1}\big) \right| \right)^2 \mathrm{d}X$$

$$\leqslant 3 \max_s \left| \hat{g}_K\big(\hat{X}_{L-n+2:L}, \hat{X}_{s-n+1:s-1}\big) - g_K\big(\hat{X}_{L-n+2:L}, \hat{X}_{s-n+1:s-1}\big) \right|^2$$

$$+ 3 \max_s \left| g_K\big(\hat{X}_{L-n+2:L}, \hat{X}_{s-n+1:s-1}\big) - g_K\big(X_{L-n+2:L}, X_{s-n+1:s-1}\big) \right|^2$$

$$+ 3 \left( \frac{(L-1-n)C_\alpha}{K^\alpha} \right)^2$$

$$\leqslant 3 \left( (2C_g^\infty + 1) C_\alpha \epsilon_{\mathsf{FFN}}^{(1)} \right)^2 + 3 \left( 2C_g^\infty C_g^{\mathrm{Lip}} C_\alpha \epsilon_{\mathsf{SA}}^{(1)} \right)^2 + 3 \left( \frac{(L-1-n)C_\alpha}{K^\alpha} \right)^2$$

$$\leqslant 3 \left( (2C_g^\infty + 1) C_\alpha \epsilon_{\mathsf{FFN}}^{(1)} + 2C_g^\infty C_g^{\mathrm{Lip}} C_\alpha \epsilon_{\mathsf{SA}}^{(1)} + \frac{(L-1-n)C_\alpha}{K^\alpha} \right)^2.$$

This estimate implies that

$$\|\mathsf{GIH}_n - \mathsf{TF}\|_{L,2}$$

$$\leqslant \sqrt{3}\left(2C_g^\infty C_g^{\mathrm{Lip}}C_\alpha \epsilon_{\mathsf{SA}}^{(1)} + (2C_g^\infty + 1)C_\alpha \epsilon_{\mathsf{FFN}}^{(1)} + \frac{(L-1-n)C_\alpha}{K^\alpha}\right) \tag{15}$$

$$\leqslant \mathcal{O}\left(A_{g,\alpha}C_2\left(\frac{C_1 nq^2}{H}\right)^q\right) + \mathcal{O}\left(\frac{B_{g,\alpha}\sqrt{K\log M}}{\sqrt{M}}\right) + \mathcal{O}\left(\frac{LC_\alpha}{K^\alpha}\right),$$

where

$$A_{g,\alpha} := C_g^\infty C_g^{\mathrm{Lip}}C_\alpha, \quad B_{g,\alpha} := (2C_g^\infty + 1)C_\alpha.$$

**Step IV. Optimizing $K$ in** (15)**.**

Notice that in RHS of (15), only $\mathcal{O}\left(\frac{C_{g,\alpha}\sqrt{K\log M}}{\sqrt{M}}\right)$ and $\mathcal{O}\left(\frac{LC_\alpha}{K^\alpha}\right)$ depend on $K$.

By Young's inequality, with $p = \frac{\alpha + \frac{1}{2}}{\alpha}$ and $q = 2(\alpha + \frac{1}{2})$, we have:

$$\min_K : \frac{\alpha}{\frac{1}{2}+\alpha}\frac{B_{g,\alpha}\sqrt{K\log M}}{\sqrt{M}} + \frac{\frac{1}{2}}{\frac{1}{2}+\alpha}\frac{LC_\alpha}{K^\alpha}$$

$$=\min_K : \frac{\alpha}{\frac{1}{2}+\alpha}\left(\left(\frac{B_{g,\alpha}\sqrt{K\log M}}{\sqrt{M}}\right)^{\frac{\alpha}{\frac{1}{2}+\alpha}}\right)^{\frac{\frac{1}{2}+\alpha}{\alpha}} + \frac{\frac{1}{2}}{\frac{1}{2}+\alpha}\left(\left(\frac{LC_\alpha}{K^\alpha}\right)^{\frac{\frac{1}{2}}{\frac{1}{2}+\alpha}}\right)^{2(\frac{1}{2}+\alpha)}$$

$$=\frac{B'_{g,\alpha}L^{1/(1+2\alpha)}}{(M/\log M)^{\alpha/(1+2\alpha)}},$$

where $B'_{g,\alpha}$ only depends on the properties of $g$ and $\alpha$.

Thus, we obtain our final bound:

$$\|\mathsf{GIH}_n - \mathsf{TF}\|_{L,2} \leqslant \mathcal{O}\left(A_{g,q}C_2\left(\frac{C_1 nq^2}{H}\right)^q\right) + \left\{\mathcal{O}\left(\frac{B_{g,\alpha}\sqrt{K\log M}}{\sqrt{M}}\right) + \mathcal{O}\left(\frac{LC_\alpha}{K^\alpha}\right)\right\}_{\min:K}$$

$$\leqslant \mathcal{O}\left(A_{g,q}\left(\frac{C_1 nq^2}{H}\right)^q\right) + \mathcal{O}\left(\frac{B'_{g,\alpha}L^{1/(1+2\alpha)}}{(M/\log M)^{\alpha/(1+2\alpha)}}\right)$$

$$\leqslant A'_{g,q}\left(\frac{C_1 nq^2}{H}\right)^q + B''_{g,\alpha}\frac{L^{1/(1+2\alpha)}}{M^{\alpha/(1+3\alpha)}},$$

where $A'_{g,\alpha}, B''_{g,\alpha}$ only depends on the properties of $g$ and $\alpha$.

For **the remained case**, $H < C_1 C_2 nq^2$, similar to the proof of Theorem 3.3 we can simply choose $\mathsf{TF}$ with all $0$ parameters. Then the approximation error can be trivially bounded by:

$$\|\mathsf{GIH}_n - \mathsf{TF}\|_{L,2} = \|\mathsf{GIH}_n - 0\|_{L,2} \leqslant 1.$$

Then, the two cases can be unified by:

$$\|\mathsf{GIH}_n - \mathsf{TF}\|_{L,2} \leqslant \begin{cases} A'_{g,\alpha}\left(\frac{C_1 nq^2}{H}\right)^q + B''_{g,\alpha}\frac{L^{1/(1+2\alpha)}}{M^{\alpha/(1+3\alpha)}}, & H \geqslant C_1 C_2 nq^2 \\ 1, & \text{otherwise} \end{cases}$$

$$\leqslant \max\{A'_{g,\alpha}, 1\}\left(\frac{C_1 \max\{C_2, 1\}nq^2}{H}\right)^q + B''_{g,\alpha}\frac{L^{1/(1+2\alpha)}}{M^{\alpha/(1+3\alpha)}}.$$

Finally, we can choose $A_{g,\alpha}'' = \max\{A_{g,\alpha}', 1\}$ and $C_{n,q} = C_1 \max\{C_2, 1\} nq^2$. Then

$$\|\mathsf{GIH}_n - \mathsf{TF}\|_{L,2} \leqslant A_{g,\alpha}'' \left(\frac{C_{n,q}}{H}\right)^q + B_{g,\alpha}'' \frac{L^{1/(1+2\alpha)}}{M^{\alpha/(1+3\alpha)}},$$

where $A_{g,\alpha}''$ and $B_{g,\alpha}''$ only depend on $g$ and $\alpha$, thus only depending on the properties of $g$. Moreover, $C_{n,q} = \mathcal{O}(nq^2)$. □

# D  PROOFS IN SECTION 4

## D.1  REPARAMETERIZATION

Despite the simplification, the transformer above is still too complicated for dynamics analysis. To overcome this challenge, we adopt the reparametrization trick used in previous works (Tian et al., 2023; Huang et al., 2023; Chen et al., 2024b). Specifically, by Theorem 3.1 and its proof, *the first layer does not require DP, and the second layer does not require RPE*. Moreover, to express the 4-gram component $f_{\mathsf{G}_4}^\star$, we only need an additional head without DP in the second layer. Therefore, we can reparameterize the model as follows:

- **The first layer.** This layer has only one trainable parameter $p^{(1,1)}$. In the unique head $\mathsf{SA}^{(1,1)}$, DP is removed by setting $W_Q^{(1,1)} = W_K^{(1,1)} = 0$, and we let $W_V^{(1,1)} = \begin{pmatrix} 0 & 0 \\ 1 & 0 \end{pmatrix}$. The output sequence of this layer given by $X^{(1)} = X^{(0)} + \mathsf{SA}^{(1,1)}(X^{(0)}) = \begin{pmatrix} x_1, \cdots, x_L \\ y_1, \cdots, y_L \end{pmatrix}$, where

$$y_s = \sum_{\tau=1}^{s-1} x_\tau \, \mathrm{sm}\left(\left(-p^{(1,1)}(s-1-\nu)\right)_{\nu=1}^{s-1}\right)_{\nu=\tau} \tag{16}$$

  for $s \in [L]$, where $p^{(1,1)}$, used in RPE, is the unique trainable parameter in this layer.

- **The second layer.** This layer has 5 trainable parameters: $w_V^{(2,1)}, w_V^{(2,2)}, p^{(2,1)}, w_K^{(2,2)}, w_Q^{(2,2)}$ for parametrizing the two heads. The first head $\mathsf{SA}^{(2,1)}$ without DP is responsible to fit $f_{\mathsf{G}_4}^\star$, while the second head $\mathsf{SA}^{(2,2)}$ without RPE is responsible to fit $f_{\mathsf{IH}_2}^\star$. Specifically, $W_Q^{(2,1)} = W_K^{(2,1)} = 0, W_V^{(2,1)} = \begin{pmatrix} 0 & w_V^{(2,1)} \\ 0 & 0 \end{pmatrix}, p^{(2,2)} = 0, W_V^{(2,2)} = \begin{pmatrix} w_V^{(2,2)} & 0 \\ 0 & 0 \end{pmatrix}$. Then the second layer processes $X^{(1)}$ and outputs the last token:

$$\mathsf{TF}_{-1}(X;\theta) = \left(\sum_{s=2}^{L-2} w_V^{(2,1)} y_s \pi_s, \sum_{s=2}^{L-2} w_V^{(2,2)} x_s \rho_s\right)^\top, \tag{17}$$

$$\pi_s = \mathrm{sm}\left(\left(-p^{(2,1)}(L-1-\nu)\right)_{\nu=2}^{L-2}\right)_{\nu=s}, \quad \rho_s = \mathrm{sm}\left(\left(x_L w_Q^{(2,2)} w_K^{(2,2)} x_{\nu-1}\right)_{\nu=2}^{L-2}\right)_{\nu=s},$$

  where $y_s$ is given by Eq. (16). $p^{(2,1)}, w_V^{(2,1)}$ are trainable parameters in $\mathsf{SA}^{(2,1)}$, while $w_Q^{(2,2)}, w_K^{(2,2)}, w_V^{(2,2)}$ are trainable parameters in $\mathsf{SA}^{(2,2)}$.

The set of all six trainable parameters across both layers is denoted by $\theta$.

## D.2 OPTIMIZATION DYNAMICS IN TRAINING STAGE I

In this subsection we focus on training the first layer of Transformer model to capture the token ahead. For simplicity, we introduce some notations:

$$\tilde{p} := p^{(1,1)}, \quad p := p^{(2,1)}, \quad g := w_V^{(2,1)}, \quad h := w_V^{(2,2)}, \quad w_K := w_K^{(2,2)}, \quad w_Q := w_Q^{(2,2)},$$

and denote the initialization of each parameter as $\tilde{p}(0), p(0), g(0), w_Q(0), w_K(0), h(0)$ respectively.

We initialize $p(0), w_k(0), w_Q(0) = 0$ while the other parameters are all initialized at $\sigma_{\text{init}}$. In this training stage, we only train $\tilde{p}$. And our goal, **the proof of Lemma 4.3** can be deduced from which, is to prove:

$$\lim_{t \to +\infty} \tilde{p}(t) = +\infty.$$

In this stage, the $s$-th output token of the first layer is represented as

$$\begin{pmatrix} x_s \\ (x_\tau)_{\tau=1}^{s-1} \text{ softmax} \left( \left( -\tilde{p}(s-1-\tau) \right)_{\tau=1}^{s-1}^\top \right) \end{pmatrix},$$

and the target function and output of transformer are as follows

$$f^*(X) = \begin{pmatrix} \frac{\alpha^\star}{1+\alpha^\star} x_{L-2} \\ \frac{1}{1+\alpha^\star} (x_s)_{s=2}^{L-1} \text{ softmax} \left( \left( x_L w^{\star 2} x_{s-1} \right)_{s=2}^{L-1}^\top \right) \end{pmatrix},$$

$$f_\theta(X) = \begin{pmatrix} g(0) \left( (x_\tau)_{\tau=1}^{s-1} \text{softmax} \left( (-\tilde{p}(s-1-\tau))_{\tau=1}^{s-1}^\top \right) \right)_{s=2}^{L-1} \text{softmax} \left( (-p(0)(L-1-s))_{s=2}^{L-1}^\top \right) \\ h(0)(x_s)_{s=2}^{L-2} \text{softmax} \left( \left( w_K(0)w_Q(0)x_L \cdot (x_\tau)_{\tau=1}^{s-1} \text{softmax} \left( (-\tilde{p}(s-1-\tau))_{\tau=1}^{s-1}^\top \right) \right)_{s=2}^{L-2}^\top \right) \end{pmatrix}$$

$$= \begin{pmatrix} g(0) \frac{1}{L-2} \sum_{\tau=1}^{L-2} \left( \sum_{s=\tau+1}^{L-1} \text{softmax} \left( (-\tilde{p}(s-1-t))_{t=1}^{s-1} \right) \right)_{t=\tau} x_\tau \\ h(0) \frac{1}{L-2} \sum_{s=2}^{L-2} x_s \end{pmatrix}.$$

Since we only focus on $\tilde{p}$ and the other parameters remain the initialization value, the loss function can be simplified as

$$\mathcal{L}(\theta) = \mathop{\mathbb{E}}_{X \sim \mathbb{N}(0,1)^L} \left[ \frac{\alpha^{\star 2}}{(1+\alpha^\star)^2} x_{L-2}^2 + \frac{g(0)^2}{(L-2)^2} \sum_{\tau=1}^{L-2} \left( \sum_{s=\tau+1}^{L-1} \text{softmax} \left( (-\tilde{p}(s-1-t))_{t=1}^{s-1} \right)_{t=\tau} \right)^2 x_\tau^2 \right.$$

$$\left. + \frac{2g(0)}{L-2} \frac{\alpha^\star}{1+\alpha^\star} \text{softmax} \left( (-p(0)(L-1-s))_{s=2}^{L-1} \right)_{s=L-1} x_{L-2}^2 \right] + C(w^\star, \alpha^\star, w(0), h(0))$$

where the second term $C(w^\star, \alpha^\star, w(0), h(0))$ is a constant depends on $w^\star, \alpha^\star, w(0)$ and $h(0)$, produced by calculating the error of the second head, i.e., loss of induction head, while the first term is 4-gram loss.

We first define several functions that will be useful for calculation in this stage and the second one:

*Function I.* This function is purely defined for the calculation of $\frac{\mathrm{d}p}{\mathrm{d}t}$. Denoted by $q(\tilde{p}) := \sum_{\tau=1}^{L-2} \left( \sum_{s=\tau+1}^{L-1} \frac{e^{\tilde{p}(s-1-\tau)}}{\sum_{k=0}^{s-2} e^{-\tilde{p}k}} \right)^2$, we first prove $\frac{\mathrm{d}q}{\mathrm{d}\tilde{p}} \leqslant 0$.

$$q(\tilde{p}) := \sum_{\tau=1}^{L-2} \left( \sum_{s=\tau+1}^{L-1} \frac{e^{\tilde{p}(s-1-\tau)}}{\sum_{k=0}^{s-2} e^{-\tilde{p}k}} \right)^2$$

$$= \sum_{\tau=1}^{L-2} \left( \sum_{s=\tau+1}^{L-1} \frac{e^{-\tilde{p}(s-1-\tau)}}{1-e^{-\tilde{p}(s-1)}} (1-e^{-\tilde{p}}) \right)^2$$

$$= (1 - e^{-\tilde{p}})^2 \sum_{\tau=1}^{L-2} \left( \sum_{s=\tau+1}^{L-1} \frac{e^{-\tilde{p}(s-1-\tau)}}{1 - e^{-\tilde{p}(s-1)}} \right)^2$$

$$= (1 - e^{-\tilde{p}})^2 \sum_{\tau=1}^{L-2} e^{2\tilde{p}\tau} \left( \sum_{s=\tau+1}^{L-1} \frac{e^{-\tilde{p}(s-1)}}{1 - e^{-\tilde{p}(s-1)}} \right)^2$$

$$= (1 - e^{-\tilde{p}})^2 \sum_{\tau=1}^{L-2} e^{2\tilde{p}\tau} \left( \sum_{s=\tau+1}^{L-1} \frac{1}{e^{\tilde{p}(s-1)} - 1} \right)^2$$

Then we take its derivative of $\tilde{p}$

$$\frac{dq}{d\tilde{p}} = 2(1 - e^{-\tilde{p}})e^{-\tilde{p}} \sum_{\tau=1}^{L-2} e^{2\tilde{p}\tau} \left( \sum_{s=\tau+1}^{L-1} \frac{1}{e^{\tilde{p}(s-1)} - 1} \right)^2$$

$$+ (1 - e^{-\tilde{p}})^2 \sum_{\tau=1}^{L-2} 2\tau e^{2\tilde{p}\tau} \left( \sum_{s=\tau+1}^{L-1} \frac{1}{e^{\tilde{p}(s-1)} - 1} \right)^2$$

$$+ (1 - e^{-\tilde{p}})^2 \sum_{\tau=1}^{L-2} 2 e^{2\tilde{p}\tau} \left( \sum_{s=\tau+1}^{L-1} \frac{1}{e^{\tilde{p}(s-1)} - 1} \right) \left( \sum_{s=\tau+1}^{L-1} \frac{-(s-1)e^{\tilde{p}(s-1)}}{(e^{\tilde{p}(s-1)} - 1)^2} \right)$$

$$= 2(1 - e^{-\tilde{p}}) \sum_{\tau=1}^{L-2} e^{2\tilde{p}\tau} \left( \sum_{s=\tau+1}^{L-1} \frac{1}{e^{\tilde{p}(s-1)} - 1} \right) \left( \sum_{s=\tau+1}^{L-1} \frac{e^{-\tilde{p}} + \tau(1 - e^{-\tilde{p}})}{e^{\tilde{p}(s-1)} - 1} - \frac{(s-1)e^{\tilde{p}(s-1)}}{(e^{\tilde{p}(s-1)} - 1)^2} \right)$$

$\frac{dq}{d\tilde{p}}$'s last factor can be formed as

$$\frac{\left(\tau - (\tau-1)e^{-\tilde{p}}\right)\left(e^{\tilde{p}(s-1)} - 1\right) - (s-1)e^{\tilde{p}(s-1)}}{e^{\tilde{p}(s-1)} - 1)^2}$$

$$= \frac{(\tau + 1 - s)t^{s-1} - (\tau-1)t^{s-2} - \tau + \frac{\tau-1}{t}}{e^{\tilde{p}(s-1)} - 1)^2}$$

where $t = e^{-\tilde{p}} \geqslant 1$. Since $s \geqslant \tau + 1$, $\frac{dq}{d\tilde{p}} \leqslant 0$.

*Function II.* For simplicity, we define $M(p)$ and its derivative $m(p)$:

$$M(p) := \sum_{s=2}^{L-1} \exp(-p(L-1-s)) = \sum_{s=0}^{L-3} \exp{-ps} = \frac{1 - e^{-p(L-2)}}{1 - e^{-p}},$$

$$m(p) := \sum_{s=1}^{l-3} s \exp(-ps) = \frac{e^{-p} - (L-2)e^{-p(L-2)} + (L-3)e^{-p(L-1)}}{(1 - e^{-p})^2}.$$

*Function III.* The third function is derivative of softmax. By straightfoward calculation, we obtain:

$$\frac{d}{dp}\text{softmax}\left((-p(L-1-t))_{t=2}^{L-1}\right)_{t=L-1-s} = \frac{d}{dp}\frac{\exp(-ps)}{\sum_{\tau=0}^{L-3} \exp(-p\tau)} = \frac{-s\exp(-ps)M(p) + \exp(-ps)m(p)}{M(p)^2}.$$

Through the quantities and their properties above, we obtain the dynamic of $\tilde{p}$

$$\frac{d\tilde{p}}{dt} = -\frac{g(0)^2}{(L-2)^2}q'(\tilde{p}) + \frac{2\alpha^\star g(0)}{(1 + \alpha^\star)(L-2)}\frac{m(p)}{M(p)^2}$$

$$\geqslant \frac{2\alpha^\star g(0)}{(1 + \alpha^\star)(L-2)}e^{-\tilde{p}},$$

which implies:

$$\lim_{t \to +\infty} \tilde{p}(t) = +\infty.$$

### D.3 OPTIMIZATION DYNAMICS IN TRAINING STAGE II

In this training stage, the first layer is already capable of capturing the token ahead i.e. $y_s = x_{s-1}$. And we train the parameters $w_{V_1}, w_{V_2}, p, w_{KQ}$ in the second layer.

We start from proving the parameter balance lemma:

**Lemma D.1** (Restate of Lemma 4.4). *In Training Stage II, it holds that* ${w_Q^{(2,2)}}^2(t) \equiv {w_K^{(2,2)}}^2(t)$.

*Proof.* Notice that

$$\frac{\mathrm{d}}{2\mathrm{d}t}\left({w_Q^{(2,2)}}^2(t) - {w_K^{(2,2)}}^2(t)\right) = -w_Q^{(2,2)}\frac{\partial\mathcal{L}}{\partial w_Q^{(2,2)}} + w_K^{(2,2)}\frac{\partial\mathcal{L}}{\partial w_K^{(2,2)}}$$

$$= -w_Q^{(2,2)}w_K^{(2,2)}\frac{\partial\mathcal{L}}{\partial\left(w_Q^{(2,2)}w_K^{(2,2)}\right)} + w_K^{(2,2)}w_Q^{(2,2)}\frac{\partial\mathcal{L}}{\partial\left(w_Q^{(2,2)}w_K^{(2,2)}\right)} \equiv 0.$$

Thus, we have:

$$ {w_Q^{(2,2)}}^2(t) - {w_K^{(2,2)}}^2(t) \equiv {w_Q^{(2,2)}}^2(0) - {w_K^{(2,2)}}^2(0) = 0.$$

$\square$

For simplicity, we still use the following notations:

$$p := p_1, \quad g := w_{V_1}, \quad w := w_{KQ}, \quad h := w_{V_2}.$$

and notations for initialization $p(0), g(0), w(0), h(0)$. Then the target function and output of Transformer can be formed as follows

$$f^\star(X) = \begin{pmatrix} \frac{\alpha^\star}{1+\alpha^\star}x_{L-2} \\ \frac{1}{1+\alpha^\star}(x_s)_{s=2}^{L-1}\mathrm{softmax}\left(\left({w^\star}^2 x_L x_{s-1}\right)_{s=2}^{L-1}\right)^\top \end{pmatrix},$$

$$\mathsf{TF}(X;\theta) = \begin{pmatrix} g\cdot(x_{s-1})_{s=2}^{L-1}\mathrm{softmax}\left((-p(L-1-s))_{s=2}^{L-1}\right)^\top \\ h\cdot(x_s)_{s=2}^{L-1}\mathrm{softmax}\left(\left(w^2 x_L x_{s-1}\right)_{s=2}^{L-1}\right)^\top \end{pmatrix}.$$

And the loss function is expressed as:

$$\mathcal{L}(\theta) = \frac{1}{2}\mathop{\mathbb{E}}_{X\sim\mathbb{N}(0,1)^L}\left[\|f^\star(x) - \mathsf{TF}(x;\theta)\|^2\right]$$

$$= \frac{1}{2}\mathbb{E}_X\left[\left(\frac{\alpha^\star}{1+\alpha^\star}x_{L-2} - g\cdot(x_{s-1})_{s=2}^{L-1}\mathrm{softmax}\left((-p(L-1-s))_{s=2}^{L-1}\right)^\top\right)^2\right]$$

$$+ \frac{1}{2}\mathbb{E}_X\left[\left(\frac{1}{1+\alpha^\star}(x_s)_{s=2}^{L-1}\mathrm{softmax}\left(\left({w^\star}^2 x_L x_{s-1}\right)_{s=2}^{L-1}\right)^\top - h\cdot(x_s)_{s=2}^{L-1}\mathrm{softmax}\left(\left(w^2 x_L x_{s-1}\right)_{s=2}^{L-1}\right)^\top\right)^2\right].$$

The total loss can naturally be divided into two parts:

$$\mathcal{L}(\theta) = \mathcal{L}_{\mathsf{G}_4}(\theta) + \mathcal{L}_{\mathsf{IH}_2}(\theta),$$

where

$$\mathcal{L}_{\mathsf{G}_4}(\theta) = \mathcal{L}_{\mathsf{G}_4}(p,g)$$

$$= \frac{1}{2}\mathbb{E}_X\left[\left(\frac{\alpha^\star}{1+\alpha^\star}x_{L-2} - g\cdot(x_{s-1})_{s=2}^{L-1}\mathrm{softmax}\left((-p(L-1-s))_{s=2}^{L-1}\right)\right)^\top)^2\right],$$

$$\mathcal{L}_{\mathsf{IH}_2}(\theta) = \mathcal{L}_{\mathsf{IH}_2}(w,h)$$

$$= \frac{1}{2}\mathbb{E}_X\left[\left(\frac{1}{1+\alpha^\star}(x_s)_{s=2}^{L-1}\mathrm{softmax}\left(\left(w^{\star 2}x_L x_{s-1}\right)_{s=2}^{L-1}\right)^\top - h\cdot(x_s)_{s=2}^{L-2}\mathrm{softmax}\left(\left(w^2 x_L x_{s-1}\right)_{s=2}^{L-1}\right)^\top\right)^2\right].$$

Notably, the dynamics of $(p, g)$ and $(w, h)$ are **decoupled**, which allows us to analyze them separately.

Additionally, we denote the optimal values of the parameters as:

$$p^\star = +\infty, \quad g^\star = \frac{\alpha^\star}{1+\alpha^\star}, \quad w^\star := w^\star, \quad h^\star = \frac{1}{1+\alpha^\star}.$$

For the initialization scale and the sequence length, we consider the case:

$$\sigma_{\mathrm{init}} = \mathcal{O}(1) \ll 1, \quad L = \Omega(1/\sigma_{\mathrm{init}}) \gg 1.$$

### D.3.1 DYNAMICS OF THE PARAMETERS FOR 4-GRAM

First, we define two useful auxiliary functions:

$$M(p) := \frac{1 - e^{-p(L-2)}}{1 - e^{-p}},$$

$$m(p) := \frac{e^{-p} - (L-2)e^{-p(L-2)} + (L-3)e^{-p(L-1)}}{(1-e^{-p})^2}.$$

Then, a straightforward calculation, combined with Lemma E.3 and Lemma E.4, yields the explicit formulation of $\mathcal{L}_{\mathsf{G}_4}(\theta)$ and the GF dynamics of $p$ and $g$:

$$\mathcal{L}_{\mathsf{G}_4}(\theta) = \frac{1}{2}\left(\frac{\alpha^\star}{1+\alpha^\star}\right)^2 + \frac{1}{2}g^2\frac{M(2p)}{M(p)^2} - \frac{\alpha^\star g}{1+\alpha^\star}\frac{1}{M(p)}. \tag{18}$$

$$\frac{\mathrm{d}p}{\mathrm{d}t} = -\frac{\partial \mathcal{L}}{\partial p} = -\frac{\partial \mathcal{L}_{\mathsf{G}_4}}{\partial p} = \frac{m(p)}{M(p)^2}\left[g^2\frac{m(2p)}{m(p)} - g^2\frac{M(2p)}{M(p)} + \frac{\alpha^\star g}{1+\alpha^\star}\right],$$

$$\frac{\mathrm{d}g}{\mathrm{d}t} = -\frac{\partial \mathcal{L}}{\partial g} = -\frac{\partial \mathcal{L}_{\mathsf{G}_4}}{\partial g} = \frac{\alpha^\star}{1+\alpha^\star}\frac{1}{M(p)} - g\frac{M(2p)}{M(p)^2},$$

Equivalently, the dynamics can be written as:

$$\frac{\mathrm{d}p}{\mathrm{d}t} = \frac{m(p)g}{M(p)^2}\left(g^\star - g\frac{M(2p)}{M(p)} + g\frac{m(2p)}{m(p)}\right),$$

$$\frac{\mathrm{d}g}{\mathrm{d}t} = \frac{1}{M(p)}\left(g^\star - g\frac{M(2p)}{M(p)}\right).$$

Notice that at the initialization, it holds that $\frac{\mathrm{d}p}{\mathrm{d}t}|_{t=0} > 0$ and $\frac{\mathrm{d}g}{\mathrm{d}t}|_{t=0} > 0$. Then we first define a hitting time:

$$T_1^g := \inf\{t > 0 : g(t) > g^\star\}.$$

Noticing $g(0) = \sigma_{\mathrm{init}} \ll g^\star$ and the continuity, $T_1^g > 0$.

Our subsequent proof can be divided into **two phases**: a monotonic phase $t < T_1^g$, and a stable convergence phase $t > T_1^g$.

**Part I. Analysis for the monotonic phase $t < T_1^g$.**

$$\frac{\mathrm{d}p}{\mathrm{d}t} = \frac{m(p)g}{M(p)^2}\left(g^\star - g\frac{M(2p)}{M(p)} + g\frac{m(2p)}{m(p)}\right) = \frac{m(p)g}{M(p)^2}\left(g^\star - g\frac{1+e^{-p(L-2)}}{1+e^{-p}} + g\frac{m(2p)}{m(p)}\right),$$

$$\frac{\mathrm{d}g}{\mathrm{d}t} = \frac{1}{M(p)}\left(g^\star - g\frac{M(2p)}{M(p)}\right) = \frac{1}{M(p)}\left(g^\star - g\frac{1+e^{-p(L-2)}}{1+e^{-p}}\right).$$

It is easy to see that $p, g$ are monotonically increasing for $t < T_1^g$. We can choose sufficiently large

$$L = \Omega(1/p(0)) = \Omega(1/\sigma_{\text{init}})$$

such that:

$$(L-3)e^{-(L-3)p(t)}, \ e^{-(L-5)p(t)} < 0.0001, \quad \forall p > \sigma_{\text{init}}.$$

Then we can calculate the following three terms in the dynamics:

$$\frac{m(p)}{M^2(p)} = \frac{e^{-p}\left(1 - (L-2)e^{-p(L-3)} + (L-3)e^{-p(L-2)}\right)}{1 - e^{-p(L-2)}} = \frac{e^{-p}(1 + \xi_1(p))}{1 + \xi_2(p)},$$

$$\frac{1}{M(p)} = \frac{1 - e^{-p(L-2)}}{1 - e^{-p}} = \frac{1 + \xi_3(p)}{1 - e^{-p}},$$

$$\frac{m(2p)}{m(p)} = \frac{e^{-p}\left(1 - (L-2)e^{-2p(L-3)} + (L-3)e^{-2p(L-2)}\right)}{(1 + e^{-p})^2\left(1 - (L-2)e^{-p(L-3)} + (L-3)e^{-p(L-2)}\right)}$$

$$= \frac{e^{-p}(1 + \xi_4(p))}{(1 + e^{-p})^2(1 + \xi_5(p))},$$

where the error functions satisfy:

$$|\xi_1(p)|, \cdots, |\xi_5(p)| \leqslant 0.0001, \ \forall t > T_1^g.$$

Then the dynamics satisfy:

$$\frac{\mathrm{d}p}{\mathrm{d}t} = \frac{e^{-p}g(1 + \xi_1(p))}{1 + \xi_2(p)}\left(g^\star - g\frac{1 + e^{-p(L-2)}}{1 + e^{-p}} + \frac{ge^{-p}(1 + \xi_3(t))}{(1 + e^{-p})^2(1 + \xi_5(t))}\right),$$

$$\frac{\mathrm{d}g}{\mathrm{d}t} = \frac{1 + \xi_3(p)}{1 - e^{-p}}\left(g^\star - g\frac{1 + e^{-p(L-2)}}{1 + e^{-p}}\right).$$

When $g < \frac{1}{2}\frac{\alpha^\star}{1+\alpha^\star}$, we have

$$\frac{\mathrm{d}p}{\mathrm{d}g} \leqslant 2\left(e^{-p} - e^{-2p}\right)g.$$

By define $T_{1/2}^g := \inf\{t > 0 : g(t) > g^\star/2\}$ and $\tilde{p} := p(T_{1/2}^g)$, we have

$$\ln(e^{\tilde{p}} - 1) \leqslant \frac{1}{4}g^{\star 2} - g(0)^2 + e^{p(0)} - 1 + \ln(e^{p(0)} - 1)$$

then $\tilde{p} \leqslant \mathcal{O}(\sqrt{p(0)})$, from which we infer that p barely increases when $t \leqslant T_{1/2}^g$.

For $0 \leqslant t \leqslant T_{1/2}^g$,

$$\frac{\mathrm{d}g}{\mathrm{d}t} \geqslant \frac{1}{1 - e^{-p(0)}}\left[g^\star - \frac{g}{1 + e^{-p(0)}}\right]$$

$$g \geqslant g^\star(1 + e^{-p(0)}) + \left[g(0) - g^\star(1 + e^{-p(0)})\right]\exp\left(\frac{-t}{1 - e^{-2p(0)}}\right)$$

so

$$T_{1/2}^g \leqslant (1 - e^{-2p(0)})\ln\left(\frac{g^\star(1 + e^{-p(0)}) - g(0)}{g^\star\left((1 + e^{-p(0)}) - \frac{1}{2}\right)}\right) = \mathcal{O}\left(2p(0)\right)$$

For $T_{1/2}^g \leqslant t \leqslant T_1^g$, let $p_1 := p(T_1^g)$,

$$\frac{\mathrm{d}p}{\mathrm{d}g} \leqslant 1.01e^{-p}(1 - e^{-p})g\left(1 + \frac{\frac{g}{1+e^{-p}} - \frac{g}{(1+e^{-p})^2}}{\frac{\alpha^\star}{1+\alpha^\star} - \frac{g}{1+e^{-p}}}\right)$$

$$\leqslant \frac{1.01}{4}\frac{\alpha^\star}{1+\alpha^\star}(1+e^{-p_1})$$

then

$$p_1 - p(0) \leqslant \frac{1.01}{4}\left(\frac{\alpha^\star}{1+\alpha^\star}\right)^2(1+e^{p_1}),$$

$$p_1 \leqslant \frac{1}{2\left(\frac{\alpha^\star}{1+\alpha^\star}\right)^2 - 1},$$

and we take $\alpha^\star > 1$.

Since for $T_{1/2}^g \leqslant t \leqslant T_1^g$,

$$\frac{\mathrm{d}p}{\mathrm{d}t} \leqslant 2e^{-p}g^\star\left(g^\star - \frac{1}{8}g^\star\right),$$

$$\frac{\mathrm{d}p}{\mathrm{d}t} \geqslant \frac{1}{2}e^{-p}g^\star\left(g^\star - \frac{1}{1+e^{-p_1}}g^\star\right),$$

we have

$$T_1^g - t_1 \leqslant \mathcal{O}\left((e^{2p_1} - 1)\left(\frac{1+\alpha^\star}{\alpha^\star}\right)^2\right).$$

Hence, putting the two part of time together we have

$$T_1^g \leqslant \mathcal{O}\left(p(0) + (e^{2p_1} - 1)\left(\frac{1+\alpha^\star}{\alpha^\star}\right)^2\right)$$

$$= \mathcal{O}\left(\sigma_{\text{init}} + (e^{2p_1} - 1)\left(\frac{1+\alpha^\star}{\alpha^\star}\right)^2\right) = \mathcal{O}(1). \tag{19}$$

**Part II. Analysis for the convergence phase $t > T_1^g$.**

We will prove that, in this phase, $(p, g)$ keep in a stable region, and the convergence occurs.

Recall the dynamics:

$$\frac{\mathrm{d}p}{\mathrm{d}t} = \frac{m(p)g}{M(p)^2}\left(g^\star - g\frac{1+e^{-p(L-2)}}{1+e^{-p}} + g\frac{m(2p)}{m(p)}\right),$$

$$\frac{\mathrm{d}g}{\mathrm{d}t} = \frac{1}{M(p)}\left(g^\star - g\frac{1+e^{-p(L-2)}}{1+e^{-p}}\right).$$

Using contradiction, it is easy to verify that for all $t > T_1^g$,

$$g^\star < g(t) < 2g^\star, \quad \frac{\mathrm{d}p(t)}{\mathrm{d}t} > 0,$$

which means $g$ has entered a stable region (although it is possible that $g$ is non-monotonic), while $p$ keeps increase. In fact, if $T_{2g^\star}^g := \inf\{t > 0 : g(t) = 2g^\star\}$, then $\frac{\mathrm{d}g}{\mathrm{d}t}|_{T_{2g^\star}^g} < 0$, which leads to a contradiction. If $T_0^{\mathrm{d}p/\mathrm{d}t} := \inf\{t > 0 : \frac{\mathrm{d}p(t)}{\mathrm{d}t} = 0\}$, then

$$\left(g^\star - g\frac{1+e^{-p(L-2)}}{1+e^{-p}} + g\frac{m(2p)}{m(p)}\right)\Bigg|_{T_0^{\mathrm{d}p/\mathrm{d}t}} = 0, \quad \frac{\mathrm{d}g}{\mathrm{d}t} < 0,$$

$$\frac{\mathrm{d}}{\mathrm{d}t}\left(g^\star - g\frac{1+e^{-p(L-2)}}{1+e^{-p}} + g\frac{m(2p)}{m(p)}\right) = -g'\frac{1+e^{-p(L-2)}}{1+e^{-p}} + g'\frac{m(2p)}{m(p)} > 0,$$

where the last inequality leads to a contradiction.

Thus, $p(t) > p(T_1^g) > p(0) = \sigma_{\text{init}}$ holds in this phase. Therefore, the dynamics

$$\frac{\mathrm{d}p}{\mathrm{d}t} = \frac{e^{-p}g(1 + \xi_1(p))}{1 + \xi_2(p)} \left( g^\star - g\frac{1 + e^{-p(L-2)}}{1 + e^{-p}} + \frac{ge^{-p}(1 + \xi_3(t))}{(1 + e^{-p})^2(1 + \xi_5(t))} \right),$$

$$\frac{\mathrm{d}g}{\mathrm{d}t} = \frac{1 + \xi_3(p)}{1 - e^{-p}} \left( g^\star - g\frac{1 + e^{-p(L-2)}}{1 + e^{-p}} \right),$$

also satisfy

$$|\xi_1(p)|, \cdots, |\xi_5(p)| \leqslant 0.0001, \ \forall t > T_1^g.$$

For simplicity, we consider the transform:

$$u := e^{-p}.$$

Then the dynamics of $u$ and $g$ can be written as:

$$\frac{\mathrm{d}u}{\mathrm{d}t} = -\frac{(1 + \xi_1(p))u^2g}{1 + \xi_2(p)} \left( g^\star - g\frac{1 + u^{L-2}}{1 + u} + \frac{gu(1 + \xi_4(p))}{(1 + u)^2(1 + \xi_5(p))} \right),$$

$$\frac{\mathrm{d}g}{\mathrm{d}t} = \frac{1 + \xi_3(p)}{1 - u} \left( g^\star - g\frac{1 + u^{L-2}}{1 + u} \right).$$

Notice that this dynamics are controlled by high-order terms. Consequently, we construct a variable to reflect the dynamics of high-order term:

$$v := ug^\star + (g^\star - g).$$

Then the dynamics of $u$ and $v$ satisfy:

$$\frac{\mathrm{d}u}{\mathrm{d}t} = -\frac{(1 + \xi_1(p))u^2g}{1 + \xi_2(p)} \left( \frac{v - u^{L-2}g}{1 + u} + \frac{gu(1 + \xi_4(p))}{(1 + u)^2(1 + \xi_5(p))} \right),$$

$$\frac{\mathrm{d}v}{\mathrm{d}t} = -\frac{(1 + \xi_1(p))u^2gg^\star}{1 + \xi_2(p)} \left( \frac{v - u^{L-2}g}{1 + u} + \frac{gu(1 + \xi_4(p))}{(1 + u)^2(1 + \xi_5(p))} \right) - \frac{1 + \xi_3(p)}{1 - u^2} \left( v - u^{L-2}g \right).$$

Now we consider the Lyapunov function about $u, v$:

$$G(u, v) := \frac{1}{2} \left( u^2 + v^2 \right).$$

Then it is straightforward:

$$\begin{aligned}
\frac{\mathrm{d}G}{2\mathrm{d}t} &= u\frac{\mathrm{d}u}{\mathrm{d}t} + v\frac{\mathrm{d}v}{\mathrm{d}t} \\
&= -\frac{u^3g(1 + \xi_1(p))}{1 + \xi_2(p)} \left( \frac{v - u^{L-2}g}{1 + u} + \frac{gu(1 + \xi_4(p))}{(1 + u)^2(1 + \xi_5(p))} \right) \\
&\quad - \frac{(1 + \xi_1(p))u^2vgg^\star}{1 + \xi_2(p)} \left( \frac{v - u^{L-2}g}{1 + u} + \frac{gu(1 + \xi_4(p))}{(1 + u)^2(1 + \xi_5(p))} \right) \\
&\quad - \frac{1 + \xi_3(p)}{1 - u^2} \left( v - u^{L-2}g \right) v.
\end{aligned}$$

By $|\xi_1|, \cdots, |\xi_5| \leqslant 0.0001$, we have the following estimate for the Lyapunov dynamics:

$$\begin{aligned}
\frac{\mathrm{d}G}{2\mathrm{d}t} &\leqslant \frac{1.001g}{1 + u}|u^3v| + \frac{1.0001g^2}{1 + u}u^{L+1} - \frac{0.999g^2}{(1 + u^2)}u^4 \\
&\quad - \frac{0.999gg^\star}{1 + u}u^2v^2 + \frac{1.001g^2g^\star}{1 + u}|u^Lv| + \frac{1.001g^2g^\star}{(1 + u^2)}|u^3v|
\end{aligned}$$

$$-\frac{0.999}{1-u^2}v^2 + \frac{1.001g}{1-u^2}|u^{L-2}v|$$

By $u^{L-5} = e^{-p(L-5)} < 0.0001$ and $0 < u < e^{-p(T_1^g)}$, we further have:

$$\frac{\mathrm{d}G}{2\mathrm{d}t} \leqslant \frac{1.002g}{1+u}|u^3v| - \frac{0.99g^2}{(1+u)^2}u^4 - \frac{0.999gg^\star}{1+u}u^2v^2 + \frac{1.005g^2g^\star}{(1+u)^2}|u^3v| - \frac{0.999}{1-u^2}v^2$$

$$\leqslant -\frac{0.99g^2}{(1+u)^2}u^4 - \frac{0.99gg^\star}{1+u}u^2v^2 - \frac{0.99}{1-u^2}v^2 + 1.01\left(\frac{g}{1+u} + \frac{g^2g^\star}{(1+u)^2}\right)|u^3v|.$$

By using the following inequalities:

$$\frac{g^2g^\star}{(1+u)^2}|u^3v| \leqslant \frac{1}{2}\left(\frac{1.98}{1.01}\frac{gg^\star}{1+u}u^2v^2 + \frac{1.01}{1.98}\frac{g^3g^\star}{(1+u)^3}u^4\right)$$

$$\frac{g}{1+u}|u^3v| \leqslant \frac{1}{2}\left(\frac{0.99}{1.01}(1+u)v^2 + \frac{1.01}{0.99}\frac{g^2}{(1+u)^3}u^6\right)$$

$$-\frac{1}{1-u^2} + \frac{1}{2}(1+u) < -\frac{2}{5}$$

we have

$$\frac{\mathrm{d}G}{\mathrm{d}t} \leqslant -0.99\frac{g^2}{(1+u)^2}u^4 + \frac{1.01}{3.96}\frac{g^3g^\star}{(1+u)^3}u^4 + \frac{1.01}{1.98}\frac{g^2}{(1+u)^3}u^6 - \frac{1.98}{5}v^2.$$

Since $g^\star < g < 2g^\star$, $u > 0$ for $t > T_1^g$, and $\frac{u^2}{1+u} \leqslant \frac{1}{2}$ for $0 \leqslant u \leqslant 1$, we have:

$$\frac{1}{4}\frac{g^3g^\star}{(1+u)^3} + \frac{1}{2}\frac{g^2u^2}{(1+u)^3} \leqslant \frac{g^2}{(1+u)^2}\left(\frac{g^{\star 2}}{2(1+u)} + \frac{u^2}{2(1+u)}\right)$$

$$\leqslant \frac{g^2}{(1+u)^2}\left(\frac{1}{2} + \frac{1}{4}\right) = \frac{3}{4}\frac{g^2}{(1+u)^2},$$

then

$$\frac{\mathrm{d}G(u,v)}{\mathrm{d}t} \leqslant -0.22\frac{g^2}{(1+u)^2}u^4 - \frac{2}{5}v^2$$

$$\leqslant -\frac{0.99}{16}g^{\star 2}u^4 - \frac{1.98}{5}v^2 \leqslant -\frac{g^{\star 2}}{65}G(u,v)^2,$$

which implies:

$$G(u(t),v(t)) \leqslant \frac{1}{G(u(t_1),v(t_1)) + \frac{g^{\star 2}}{64}(t-t_1)}, \quad \forall t > T_1^g.$$

Hence,

$$u^2(t), \quad v^2(t) = \mathcal{O}\left(\frac{1}{g^{\star 2}t}\right) = \mathcal{O}\left(\frac{1}{t}\right), \quad \forall t > T_1^g = \mathcal{O}(1)$$

which implies:

$$e^{-p(t)} = u(t) = \mathcal{O}\left(\frac{1}{\sqrt{t}}\right), \quad \forall t > T_1^g = \mathcal{O}(1);$$

$$g(t) - g^\star = g^\star u(t) - v(t) \leqslant \mathcal{O}\left(\frac{g^\star}{\sqrt{t}}\right) + \mathcal{O}\left(\frac{1}{\sqrt{t}}\right) = \mathcal{O}\left(\frac{1}{\sqrt{t}}\right), \quad \forall t > T_1^g = \mathcal{O}(1). \tag{20}$$

**Notably**, these proofs capture the **entire** training dynamics of $p, g$, from $t = 0$ to $t = T_1^g$, and finally to $t \to +\infty$, providing a fine-gained analysis for each phase.

### D.3.2 DYNAMICS OF THE PARAMETERS FOR INDUCTION HEAD

Recall the partial loss about the induction head:

$$\mathcal{L}_{\mathsf{IH}_2}(\theta) = \frac{1}{2}\mathbb{E}_X\left[\left(\frac{1}{1+\alpha^\star}(x_s)_{s=2}^{L-1}\mathrm{softmax}\left(\left(w^{\star 2}x_Lx_{s-1}\right)_{s=2}^{L-1}\right)^\top - h\cdot(x_s)_{s=2}^{L-2}\mathrm{softmax}\left(\left(w^2x_Lx_{s-1}\right)_{s=2}^{L-2}\right)^\top\right)^2\right].$$

**Technical simplification.** Unlike $\mathcal{L}_{\mathsf{G}_4}(\theta)$, the denominators of the softmax terms $\mathrm{softmax}\left(\left(w^{\star 2}x_Lx_{s-1}\right)_{s=2}^{L-1}\right)$ and $\mathrm{softmax}\left(\left(w^2x_Lx_{s-1}\right)_{s=2}^{L-2}\right)$ in $\mathcal{L}_{\mathsf{IH}_2}(\theta)$ depend on the input tokens $X$, making it hard to derive a closed-form expression for $\mathcal{L}_{\mathsf{IH}_2}(\theta)$. In Bai et al. (2023), the authors consider a simplified transformer model, which replaces $\mathrm{softmax}(z_1, \cdots, z_L)$ with $\frac{1}{L}\exp(z_1, \cdots, z_L)$. This approximation is nearly tight when $z_1, \cdots, z_L \approx 0$. Notice that 1) $w^2x_Lx_{s-1} \approx 0$ holds near the small initialization, i.e., for $w \approx \sigma_{\mathrm{init}} \ll 1$. In fact, our analysis shows that $w \approx \sigma_{\mathrm{init}}$ is maintained over a long period. 2) $w^\star = \mathcal{O}(1)$, which implies that $w^2x_Lx_{s-1} \approx 0$ for most input sequence. Thus, we adopt the simplification used in Bai et al. (2023), resulting in the following approximation of the loss function:

$$\mathcal{L}_{\mathsf{IH}_2}(\theta) := \frac{1}{2}\mathbb{E}_X\left[\left(\frac{1}{1+\alpha^\star}\frac{1}{L-2}\sum_{s=2}^{L-1}\exp(w^{\star 2}x_Lx_{s-1})x_s - h\frac{1}{L-2}\sum_{s=2}^{L-2}\exp(w^2x_Lx_{s-1})x_s\right)^2\right].$$

Then by a straightforward calculation with Lemma E.3, we can derive its explicit formulation:

$$\mathcal{L}_{\mathsf{IH}_2}(\theta) = \frac{(1-4w^{\star 4})^{-\frac{1}{2}}}{2(1+\alpha^\star)^2(L-2)} + \frac{1}{2}\frac{h^2}{L-2}(1-4w^4)^{-\frac{1}{2}} - \frac{h(1-(w^2+w^{\star 2})^2)^{-\frac{1}{2}}}{(1+\alpha^\star)(L-2)}. \tag{21}$$

Furthermore, we can calculate GF dynamics as follows:

$$\frac{\mathrm{d}w}{\mathrm{d}t} = \frac{h}{(1+\alpha^\star)(L-2)}(1-(w^2+w^{\star 2})^2)^{-\frac{3}{2}}\cdot(w^2+w^{\star 2})\cdot 2w - \frac{h^2}{L-2}(1-4w^4)^{-\frac{3}{2}}\cdot 4w^3,$$

$$\frac{\mathrm{d}h}{\mathrm{d}t} = \frac{1}{(1+\alpha^\star)(L-2)}(1-(w^2+w^{\star 2})^2)^{-\frac{1}{2}} - \frac{h}{L-2}(1-4w^4)^{-\frac{1}{2}}.$$

For simplicity, we denote:

$$w^\star := w^\star, \quad h^\star := \frac{1}{1+\alpha^\star}.$$

**Part I. The trend and monotonicity of $w, h$.**

For simplicity, we denote the tuning time point of $h$:

$$T_2^h := \inf\left\{t > 0 : \frac{\mathrm{d}h(t)}{\mathrm{d}t} = 0\right\}.$$

In this step, we will prove the following three claims regarding the trend and monotonicity of $w, h$, which are essential for our subsequent analysis:

- **(P1.1)** $h$ initially increases beyond $h^\star$, and then remains above this value.
- **(P1.2)** $w$ keeps increasing but always stays below $w^\star$.
- **(P1.3)** $h$ increases before $T_2^h$, but decreases after $T_2^h$.

**(P1.1)** *$h$ initially increases beyond $h^\star$, and then remains above this value.*

We will prove that initially, $h$ increases beyond $h^\star$, and keeps growing beyond $h^\star$. Define

$$T_1^h := \inf\{t > 0 : h(t) > h^\star\},$$

we will prove that $h$ remains above $h^\star$ thereafter.

For simplicity, we denote

$$\psi(x) = (1 - x^2)^{-\frac{1}{2}}, \quad \phi(x) = (1 - x^2)^{-\frac{3}{2}} \cdot x,$$

then the dynamics holds:

$$\frac{\mathrm{d}h}{\mathrm{d}t} = \frac{h}{L-2}\psi(w^2 + w^{\star 2})\left[\frac{h^\star}{h} - \frac{\psi(2w^2)}{\psi(w^2 + w^{\star 2})}\right],$$

$$\frac{\mathrm{d}w}{\mathrm{d}t} = \frac{2h^2 w}{L-2} \cdot \phi(w^2 + w^{\star 2}) \cdot \left[\frac{h^\star}{h} - \frac{\phi(2w^2)}{\phi(w^2 + w^{\star 2})}\right].$$

Notice that $\frac{\phi(2w^2)}{\phi(w^2 + w^{\star 2})} < \frac{\psi(2w^2)}{\psi(w^2 + w^{\star 2})}$, $w < w^\star$, while $\frac{\phi(2w^2)}{\phi(w^2 + w^{\star 2})} > \frac{\psi(2w^2)}{\psi(w^2 + w^{\star 2})}$, $w > w^\star$.

We denote the first hitting time of $h$ decreasing to $h^\star$ as $T_{h^\star}^h$:

$$T_{h^\star}^h := \inf\left\{t > T_2^h : h(t) < h^\star\right\}.$$

If $w(T_{h^\star}^h) \geqslant w^\star$, then at the first hitting time of $w$ increasing to $w^\star$, $\frac{\mathrm{d}w}{\mathrm{d}t} < 0$, which leads to a contradiction. If $w(T_{h^\star}^h) < w^\star$, then $\frac{\mathrm{d}h}{\mathrm{d}t}|_{T_{h^\star}^h} > 0$, which also leads to a contradiction. Hence, $T_{h^\star}^h = +\infty$, which means that $h$ always remains above $h^\star$ for $t > T_2^h$.

**(P1.2)** *$w$ keeps increasing but always below $w^\star$.*

We first prove that $w$ always remains below $w^\star$. We denote the first hitting time of $w$ increasing to $w^\star$ as $t'$, then it is not difficult to see $\frac{\mathrm{d}w}{\mathrm{d}t}|_{t'} < 0$, which leads to a contradiction.

Next we prove that $w$ keeps increasing throughout. We define the following functions

$$H := \frac{1}{1+\alpha^\star}\left(1 - (w^2 + w^{\star 2})^2\right)^{-\frac{3}{2}}(w^2 + w^{\star 2}) - h(1 - 4w^4)^{-\frac{3}{2}} \cdot 2w^2$$

$$Q := \frac{1}{1+\alpha^\star}\left(1 - (w^2 + w^{\star 2})^2\right)^{-\frac{1}{2}} - h\left(1 - 4w^4\right)^{-\frac{1}{2}}$$

If at some $\bar{t}$, $\frac{\mathrm{d}w}{\mathrm{d}t}$ reaches its zero point at the first time, then

$$\left.\frac{\mathrm{d}H}{\mathrm{d}t}\right|_{\bar{t}} = -h'(\bar{t})(1 - 4w^{\star 4})^{-\frac{3}{2}} \cdot 2w(\bar{t}) > 0,$$

which leads to a contradiction. Hence $\bar{t}$ does not exist and $w$ keeps increasing.

**(P1.3)** *After the tuning point $t > T_2^h$, $h$ will be monotonically decreasing.*

The first sign-changing zero point of $\frac{\mathrm{d}h}{\mathrm{d}t}$ is $T_2^h$, then $Q(T_2^h) = 0$. $H(T_2^h) > 0$,

$$\left.\frac{\mathrm{d}Q}{\mathrm{d}t}\right|_{T_2^h} = \frac{1}{1+\alpha^\star}(1 - (w(T_2^h)^2 + w^{\star 2})^2)^{-\frac{1}{2}} \cdot 2w(T_2^h) \cdot w'(T_2^h)$$

$$\cdot \left[(1 - (w(T_2^h)^2 + w^{\star 2})^2)^{-1} \cdot (w(T_2^h)^2 + w^{\star 2}) - (1 - 4w(T_2^h)^4)^{-1} \cdot 4w(T_2^h)^2\right].$$

We can see that $T_2^h$ is a sign-changing zero point only if

$$\frac{(1 - 4w(T_2^h)^4) \cdot (w(T_2^h)^2 + w^{\star 2})}{(1 - (w(T_2^h)^2 + w^{\star 2})^2) \cdot 4w(T_2^h)^2} < 1,$$

i.e. we have:

$$w(T_2^h) > w^\circ := \sqrt{\frac{3 - 4w^{\star 4} - \sqrt{(4w^{\star 4} - 3)^2 - 16w^{\star 4}}}{8w^{\star 2}}} \geqslant \frac{w^\star}{2}, \tag{22}$$

when $w^\star = \mathcal{O}(1)$.

Next we show that $h$ keeps decreasing after $T_2^h$. We denote the first zero point of $\frac{\mathrm{d}h}{\mathrm{d}t}$ as $t^\circ$, then $Q(t^\circ) = 0$. Since $\frac{\mathrm{d}w}{\mathrm{d}t}|_{t^\circ} > 0$, we have $\frac{\mathrm{d}Q}{\mathrm{d}t}|_{t^\circ} > 0$ which leads to a contradiction. Hence $t^\circ$ does not exist and $h$ keeps decreasing after $T_2^h$.

**Part II. Estimation of $T_1^h$, $T_2^h$, and the tight estimate of $w(t)$ before $T_2^h$.**

At the first stage, we prove that $h$ grows first and $w$ barely increases. If $w \leqslant 0.01w^\star$ and $h \leqslant \frac{1}{1+\alpha^\star} \frac{(1-w^{\star 4})^{-\frac{1}{2}}}{(1-0.01^4 w^{\star 4})^{-\frac{1}{2}}}$,

$$\frac{\mathrm{d}h}{\mathrm{d}t} \geqslant \frac{-1}{L-2}\left[ h(1-0.01^4 w^{\star 4})^{-\frac{1}{2}} - \frac{1}{1+\alpha^\star}(1-w^{\star 4})^{-\frac{1}{2}} \right],$$

$$h \geqslant \frac{1}{1+\alpha^\star}\frac{(1-w^{\star 4})^{-\frac{1}{2}}}{(1-0.01^4 w^{\star 4})^{-\frac{1}{2}}} - \left[ \frac{1}{1+\alpha^\star}\frac{(1-w^{\star 4})^{-\frac{1}{2}}}{(1-0.01^4 w^{\star 4})^{-\frac{1}{2}}} - h(0) \right] \exp\left( \frac{-t}{(L-2)(1-0.01w^{\star 4})^{\frac{1}{2}}} \right).$$
(23)

For $h$ increasing from $h(0)$ to $\frac{1}{1+\alpha^\star}$, it takes

$$T_1^h \leqslant (1-0.01w^{\star 4})^{\frac{1}{2}}(L-2)\ln\left( \frac{1}{1 - \frac{(1-w^{\star 4})^{\frac{1}{2}}}{(1-0.01^4 w^{\star 4})^{\frac{1}{2}}}} \right)$$

$$\leqslant 2(L-2)(1 - \frac{1}{2}w^{\star 4}) = \mathcal{O}(L).$$
(24)

For $0 \leqslant t \leqslant T_1^h$,

$$\frac{\mathrm{d}w}{\mathrm{d}t} \leqslant \frac{1}{L-2}(1-4w^{\star 4})^{-\frac{3}{2}} \cdot w^{\star 2} \cdot 4w.$$

Hence, it take $\mathcal{O}(L\log(1/\sigma_{\mathrm{init}}))$ for $w$ to reach $0.01w^\star$, which allows sufficient time for $h$ to reach $\frac{1}{1+\alpha^\star}$ beforehand.

Therefore, there exists a small constant $\varepsilon(w(0), w^\star)$ only depends on $w(0)$ and $w^\star$ such that $h$ is dominated by $1 + \varepsilon(w(0), w^\star)$ times right hand side of (23), from which we deduce that (24) is a tight estimation of $T_1^h$ instead of an upper bound, i.e. $T_1^h = \Theta(L)$.

We then give a bound for $h(T_2^h)$. By $\frac{\mathrm{d}h}{\mathrm{d}t} = 0$,

$$h(T_2^h)/h^\star \leqslant \frac{(1-4w^4)^{\frac{1}{2}}}{(1-(w^2+w^{\star 2})^2)^{\frac{1}{2}}} := r(w).$$

Moreover, $r(w)$ is an decreasing function of $w$ for $w > w^\circ$, and $w^\circ$ is a function of $w^\star$, we have

$$h(T_2^h)/h^\star \leqslant r(w^\circ) := R(w^\star),$$

where $w^\circ$ is a function about $w^\star$, defined in Eq. (22). It is clear that

$$R(w^\star = 0) = 1, \quad R'(w^\star = 0) = 0.$$

Then using the continuity of $R'(\cdot)$ (in $[0, 0.4]$), there exists $c > 0$ such that $|R'(w^\star)| < 0.04$ holds for all $0 < w^\star < c$, which implies:

$$R(w^\star) = R(0) + \int_0^{w^\star} R'(v)\mathrm{d}v < 1 + 0.04w^\star, \quad 0 < w^\star < c.$$

i.e., if $w^\star = O(1)$, then $R(w^\star) < 1 + 0.04w^\star$. This implies:

$$h^\star \leqslant h(t) \leqslant (1 + 0.04375w^\star)h^\star, \quad \forall t \geqslant T_1^h.$$
(25)

By some computation, we can prove that $w^\circ(w^\star)$ is an increasing function of $w^\star$, and is always above $\frac{1}{2}w^\star$. Thus we obtain a lower bound of $w^\circ$ for the estimation of lower bound of $T_2^h$:

For the second stage, $h$ barely changes and $w$ starts to grow exponentially fast, and we use the tight estimation of $T_{1/2}^w := \inf\left\{t > 0 : w(t) > \frac{1}{2}w^\star\right\}$ to give a lower bound of $T_2^h$. During this stage,

$$
\frac{\mathrm{d}w}{\mathrm{d}t} \leqslant \frac{2w}{(1+\alpha^\star)^2(L-2)}\left[(1-(w^2+w^{\star 2})^2)^{-\frac{3}{2}} \cdot (w^2+w^{\star 2}) - (1-4w^4)^{\frac{3}{2}}\right]
$$

$$
\leqslant \frac{2w}{(1+\alpha^\star)^2(L-2)}(1-4w^{\star 4})^{\frac{3}{2}} \cdot 2w^{\star 2},
$$

and $w$ has upper bound

$$
w \leqslant w(0)\exp\left(\frac{4w^{\star 2}(1-4w^{\star 4})^{\frac{3}{2}}}{(1+\alpha^\star)(L-2)}t\right). \tag{26}
$$

Hence, the lower bound of time for $w$ to reach $\frac{1}{2}w^\star$ is

$$
T_{1/2}^w - T_1^h = \frac{(1+\alpha^\star)^2(L-2)}{4w^{\star 2}(1-4w^{\star 4})^{\frac{3}{2}}}\ln(\frac{w^\star}{2w(0)}),
$$

and lower bound for $T_{1/2}^w$ is

$$
T_{1/2}^w \geqslant (L-2)\left[\frac{(1+\alpha^\star)^2\ln(\frac{w^\star}{2w(0)})}{4w^{\star 2}(1-4w^{\star 4})^{\frac{3}{2}}} - \ln\left(1-(1-w^{\star 4})^{\frac{1}{2}}\right)\right]
$$

$$
\geqslant \frac{(L-2)(1+\alpha^\star)^2}{16w^{\star 2}}\ln\left(\frac{1}{w(0)}\right) = \Omega\left(\frac{(1+\alpha^\star)^2 L}{w^{\star 2}}\log\left(\frac{1}{\sigma_{\text{init}}}\right)\right). \tag{27}
$$

On the other hand, we estimate the lower bound of $w$. Let

$$
C(x) = (1-x^2)^{-\frac{3}{2}} \cdot x,
$$

then

$$
C'(x) = 3(1-x^2)^{-\frac{5}{2}}x^2 + (1-x^2)^{-\frac{3}{2}} > 1, \quad 0 < x < 1,
$$

$$
C''(x) = 15x^3(1-x^2)^{-\frac{7}{2}} + 6x(1-x^2)^{-\frac{5}{2}} + 3x(1-x^2)^{-\frac{5}{2}} > 0, \quad 0 < x < 1.
$$

$C(x)$ is a monotonically increasing convex function on $(0,1)$ and $C(x) \geqslant x$.

Using conclusions above, before $w^2$ increases to $\frac{1}{2\gamma(w^\star)+\beta-1}w^{\star 2}$ for some $\beta > 0$,

$$
C(w^2 + w^{\star 2})
$$

$$
\geqslant C((2\gamma(w^\star) + \beta)w^2)
$$

$$
\geqslant C(2\gamma(w^\star) \cdot w^2) + C(\beta w^2) \quad \text{(Lemma E.6)}
$$

$$
\geqslant \gamma(w^\star) \cdot C(2w^2) + \beta w^2 \quad (C(ax) \geqslant aC(x), \text{ for } a > 1)
$$

then we have

$$
\frac{\mathrm{d}w}{\mathrm{d}t} \geqslant \frac{2w}{(1+\alpha^\star)^2(L-2)}(C(w^2+w^{\star 2}) - \gamma(w^\star) \cdot C(2w^2))
$$

$$
\geqslant \frac{2w}{(1+\alpha^\star)^2(L-2)}\frac{\beta}{\gamma(w^\star)+\beta}w^{\star 2}
$$

and

$$
w \geqslant w(0)\exp\left(\frac{2\beta}{\gamma(w^\star)+\beta}\frac{1}{(1+\alpha^\star)^2(L-2)}w^{\star 2}t\right).
$$

Take $\beta = 2$, then

$$
w \geqslant w(0)\exp\left(\frac{w^{\star 2}t}{(1+\alpha^\star)^2(L-2)}\right), \quad \forall t \in [0, T_{1/2}^w]. \tag{28}
$$

From the above inequality, (27) is not only an upper bound, but a tight estimation of $T_{1/2}^w$, i.e.

$$
T_{1/2}^w = \Theta\left(\frac{(1+\alpha^\star)^2 L}{w^{\star 2}}\log\left(\frac{1}{\sigma_{\text{init}}}\right)\right).
$$

**Part II. Dynamics after the critical point $T_{1/2}^w$.**

For simplicity, we consider:
$$v := w^2,$$

and denote $v^\star := w^{\star 2}, h^\star := \frac{1}{1+\alpha^\star}$. Then we focus on the dynamics of $v$ and $h$.

Additionally, we introduce a few notations used in this part:
$$\phi(x) := \frac{x}{(1-x^2)^{3/2}}, \quad \psi(x) := \frac{1}{(1-x^2)^{1/2}}.$$

Then the dynamics of $v$ and $g$ are:
$$\frac{\mathrm{d}v}{\mathrm{d}t} = \frac{4vh}{L-2}\big(h^\star\phi(v+v^\star) - h\phi(2v)\big),$$
$$\frac{\mathrm{d}h}{\mathrm{d}t} = \frac{1}{L-2}\big(h^\star\psi(v+v^\star) - h\psi(2v)\big).$$

**Step II.1.** *A coarse estimate of the relationship between $v$ and $h$.*

It is easy to verify the monotonicity that $\frac{\mathrm{d}v}{\mathrm{d}t} > 0$ and $\frac{\mathrm{d}h}{\mathrm{d}t} < 0$ for $t > t_2$. Additionally, we have
$$\frac{\psi(v+v^\star)}{\psi(2v)} < \frac{h}{h^\star} < \frac{\phi(v+v^\star)}{\phi(2v)}.$$

Then by Monotone convergence theorem, we obtain:
$$\lim_{t\to+\infty} v = v^\star, \quad \lim_{t\to+\infty} h = h^\star.$$

**Step II.2.** *Convergence analysis by Lyapunov function.*

This step aims to establish the convergence rate of $v$ and $h$.

In fact, the dynamics of $v, h$ can be approximately characterized by their linearized dynamics. In contrast, the dynamics of $p, g$ are controlled by high-order terms. Therefore, the proof for $v$ and $h$ is significantly simpler than the corresponding proof for $p$ and $g$. We only need to consider the simplest Lyapunov function:
$$G(v, h) := \frac{1}{2}\Big((v-v^\star)^2 + (h-h^\star)^2\Big).$$

It is easy to verify that
$$(L-2)\frac{\mathrm{d}G(v,h)}{\mathrm{d}t} = (v-v^\star)\frac{\mathrm{d}v}{\mathrm{d}t} + (h-h^\star)\frac{\mathrm{d}h}{\mathrm{d}t}$$
$$= 4vh(v-v^\star)\big(h^\star\phi(v+v^\star) - h\phi(2v)\big) + (h-h^\star)\big(h^\star\psi(v+v^\star) - h\psi(2v)\big)$$
$$= 4vh(v-v^\star)\Big(\phi(v+v^\star)(h^\star - h) - h(\phi(v+v^\star) - \phi(2v))\Big)$$
$$\quad + (h-h^\star)\Big((h^\star - h)\psi(v+v^\star) + h(\psi(v+v^\star) - \psi(2v))\Big)$$
$$= -4vh^2(v^\star - v)(\phi(v+v^\star) - \phi(2v)) - \psi(v+v^\star)(h-h^\star)^2$$
$$\quad + 4vh\phi(v+v^\star)(v-v^\star)(h^\star - h) + h(h-h^\star)(\psi(v+v^\star) - \psi(2v)).$$

Let $v^\star \leqslant 0.3 = \mathcal{O}(1)$. Recalling (22) and (25), as well as the monotonicity about $p$ and $w$, we have:
$$\frac{v^\star}{4} < v(t) < v^\star; \quad h^\star < h(t) < 1.02h^\star, \quad \forall t > T_2^h.$$

Combining these estimates with the properties of $\phi$ and $\psi$, we have the following straight-forward estimates:
$$\phi(v+v^\star) - \phi(2v) = \phi'(\xi)(v^\star - v) = \frac{1+2\xi^2}{(1-\xi^2)^{5/2}}(v^\star - v) \geqslant v^\star - v;$$

$$\phi(v + v^\star) \leqslant \phi(2v^\star) \leqslant 1;$$

$$\psi(v + v^\star) = \frac{1}{(1 - (v + v^\star)^2)^{1/2}} \geqslant 1;$$

$$\psi(v + v^\star) - \psi(2v) = \psi'(\xi)(v^\star - v) = \frac{\xi}{(1 - \xi^2)^{3/2}}(v^\star - v) \leqslant 1.3v^\star(v^\star - v).$$

Thus, we have the following estimate for the Lyapunov function:

$$(L - 2)\frac{\mathrm{d}G(v, h)}{\mathrm{d}t}$$

$$\leqslant -\frac{4}{1.02}v^\star h^{\star 2}(v - v^\star)^2 - (h - h^\star)^2$$

$$+ 4.08v^\star h^\star(v - v^\star)(h^\star - h) + 1.3 \cdot 1.02 v^\star h^\star(v^\star - v)(h - h^\star)$$

$$= -\frac{4}{1.02}v^\star h^{\star 2}(v - v^\star)^2 - (h - h^\star)^2 + 5.41 v^\star h^\star(v^\star - v)(h - h^\star)$$

$$\leqslant -3.92 v^\star h^{\star 2}(v - v^\star)^2 - (h - h^\star)^2 + \left(9.6 v^{\star 2}h^{\star 2}(v - v^\star)^2 + \frac{3}{4}(h - h^\star)^2\right)$$

$$\leqslant -(3.92 - 9.6 \cdot 0.3)v^\star h^{\star 2}(v - v^\star)^2 - 0.25(h - h^\star)^2 \leqslant -\frac{1}{4}v^\star h^{\star 2}G(v, h).$$

Consequently, we have the exponential bound for all $t > T_2^h$:

$$G(v(t), h(t)) \leqslant G\left(v(T_2^h), h(T_2^h)\right) \exp\left(-\frac{v^\star h^{\star 2}}{4(L - 2)}(t - T_2^h)\right), \quad \forall t > T_2^h,$$

This can imply:

$$(h(t) - h^\star)^2 = (h(T_2^h) - h^\star)^2 \exp\left(-\Omega\left(\frac{w^{\star 2}(t - T_2^h)}{L(1 + \alpha^\star)^2}\right)\right)$$

$$= \mathcal{O}\left(h^{\star 2} \exp\left(-\Omega\left(\frac{w^{\star 2}(t - T_2^h)}{L(1 + \alpha^\star)^2}\right)\right)\right), \quad \forall t > T_2^h;$$

$$(w(t) - w^\star)^2 = (w(T_2^h) - w^\star)^2 \exp\left(-\Omega\left(\frac{w^{\star 2}(t - T_2^h)}{L(1 + \alpha^\star)^2}\right)\right)$$

$$= \mathcal{O}\left(w^{\star 2} \exp\left(-\Omega\left(\frac{w^{\star 2}(t - T_2^h)}{L(1 + \alpha^\star)^2}\right)\right)\right), \quad \forall t > T_2^h. \tag{29}$$

**Notably**, these proofs capture the **entire** training dynamics of $w, h$, from $t = 0$ to $t = T_1^h$, to $t = T_{1/2}^w \leqslant T_2^h$, and finally to $t \to +\infty$, providing a fine-gained analysis for each phase.

D.4 PROOF OF THEOREM 4.5

This theorem is a direct corollary of our analysis of the entire training dynamics in Appendix D.3.1 and D.3.2, leveraging the relationship between the parameters and the loss.

*Proof of Phase I (partial learning).*

By combining (18) and (20), it follows that: $\mathcal{L}_{\mathsf{G}_4}(\theta(0)) = \Theta(1)$. Moreover,

$$\mathcal{L}_{\mathsf{G}_4}(\theta(t)) = \mathcal{O}\left(\frac{1}{t}\right), \quad t > T_1^g = O(1).$$

Thus, there exists a sufficiently large $T_{\mathrm{I}} = \Theta(1)$, such that:

$$\mathcal{L}_{\mathsf{G}_4}(\theta(T_{\mathrm{I}})) \leqslant 0.01 \mathcal{L}_{\mathsf{G}_4}(\theta(0)).$$

Recalling our proof in Appendix D.3.2, for $t < T^h_{1/2} = \mathcal{O}(L)$, it holds that $h(t) < \sigma_{\text{init}} + \mathcal{O}(t/((1+\alpha^\star)L)), w(t) < \sigma_{\text{init}} + o(t/((1+\alpha^\star)L))$. Additionally, since $T_{\text{I}} = \Theta(1) \ll \Theta(L)$, it follows that

$$w(T_{\text{I}}) = \mathcal{O}(\sigma_{\text{init}} + 1/L) < 2\sigma_{\text{init}} \ll w^\star, \quad h(T_{\text{I}}) = \mathcal{O}(\sigma_{\text{init}} + 1/L) < 2\sigma_{\text{init}} \ll h^\star.$$

Substituting these estimates into (21), we obtain by Lipschitz continuity of $\mathcal{L}_{\text{IH}_2}$:

$$|\mathcal{L}_{\text{IH}_2}(\theta(T_{\text{I}})) - \mathcal{L}_{\text{IH}_2}(\theta(0))| \leqslant 2\sigma_{\text{init}} \left( \left| \frac{\partial \mathcal{L}_{\text{IH}_2}}{\partial w} \right| + \left| \frac{\partial \mathcal{L}_{\text{IH}_2}}{\partial h} \right| \right)$$

$$\leqslant 2\sigma_{\text{init}} \left( \mathcal{O}\left( \frac{1}{(1+\alpha^\star)L} \right) + o\left( \frac{1}{(1+\alpha^\star)L} \right) \right)$$

$$\leqslant 0.01\mathcal{L}_{\text{IH}_2}(\theta(0)).$$

Thus,

$$\mathcal{L}_{\text{IH}_2}(\theta(T_{\text{I}})) \geqslant 0.99\mathcal{L}_{\text{IH}_2}(\theta(0)).$$

*Proof of Phase II (plateau) + Phase III (emergence).*

First, (26) and (28) ensures that $w$ grows exponentially before $t < T^w_{1/2}$:

$$\sigma_{\text{init}} \exp\left( \frac{w^{\star 2}}{(1+\alpha^\star)^2(L-2)} t \right) \leqslant w \leqslant \sigma_{\text{init}} \exp\left( \frac{4w^{\star 2}(1-4w^{\star 4})^{\frac{3}{2}}}{(1+\alpha^\star)(L-2)} t \right).$$

Thus, we have:

$$w(t) = \sigma_{\text{init}} \exp\left( \Theta\left( \frac{w^{\star 2}t}{(1+\alpha^\star)^2 L} \right) \right), \quad t < \Theta\left( \frac{(1+\alpha^\star)^2 L}{w^{\star 2}} \log\left( \frac{1}{\sigma_{\text{init}}} \right) \right).$$

Now we define the observation time $T_o := T^h_1 = \Theta(L)$. Notably,

$$h(T_o) = h^\star, \quad w(T_o) < 0.01w^\star.$$

The exponential growth of $w$ further implies:

$$T^w_{0.01} := \{t > 0 : w(t) > 0.01w^\star\} = \Theta\left( \frac{(1+\alpha^\star)^2 L}{w^{\star 2}} \log\left( \frac{1}{\sigma_{\text{init}}} \right) \right).$$

Regarding the dynamics of $h$, by (25), we have $|h(t) - h(T_o)| < 0.02|h(T_o)|, \forall t \geqslant T_o$.

Now we incorporate these facts ( $0 < w(T_o) < 0.01w^\star$, $0 < w(T^w_{0.01}) \leqslant 0.01w^\star$, $|h(T^w_{0.01}) - h(T_o)| < 0.02|h(T_o)|$, $h(T_o) = h^\star$) into the loss (21). By the Lipschitz continuity of $\mathcal{L}_{\text{IH}_2}$, it is straightforward that

$$\mathcal{L}_{\text{IH}_2}(\theta(T^w_{0.01})) \geqslant 0.99\mathcal{L}(\theta(T_o)).$$

Thus, we have established the lower bound for $T_{\text{II}}$:

$$T_{\text{II}} := \inf \left\{ t > T_o : \mathcal{L}_{\text{IH}_2}(\theta(t)) \leqslant 0.99 \cdot \mathcal{L}_{\text{IH}_2}(\theta(T_o)) \right\}$$

$$\geqslant T^w_{0.01} = \Omega\left( \frac{(1+\alpha^\star)^2 L}{w^{\star 2}} \log\left( \frac{1}{\sigma_{\text{init}}} \right) \right).$$

Combining the loss (21) and our parameter estimates (29), we obtain:

$$\mathcal{L}_{\text{IH}_2}(\theta(t)) = \mathcal{O}\left( \exp\left( -\Omega\left( \frac{w^{\star 2}t}{L(1+\alpha^\star)^2} \right) \right) \right), \; t > T^h_2 = \Theta\left( \frac{(1+\alpha^\star)^2 L}{w^{\star 2}} \log\left( \frac{1}{\sigma_{\text{init}}} \right) \right).$$

This implies the upper bound for $T_{\text{III}}$:

$$T_{\text{III}} := \inf \{t > T_o : \mathcal{L}_{\text{IH}_2}(\theta(t)) \leqslant 0.01 \cdot \mathcal{L}_{\text{IH}_2}(\theta(T_o))\}$$

$$= T_{1/2}^w + \mathcal{O}\left((\alpha^\star + 1)^2 L \log(1/\sigma_{\text{init}})/w^{\star 2}\right) = \mathcal{O}\left((\alpha^\star + 1)^2 L \log(1/\sigma_{\text{init}})/w^{\star 2}\right).$$

Combining the fact $T_{\text{II}} < T_{\text{III}}$, the lower bound for $T_{\text{II}}$, and the uppper bound for $T_{\text{III}}$, we obtain the two-sided bounds for both $T_{\text{II}}$ and $T_{\text{III}}$:

$$T_{\text{II}}, \; T_{\text{III}} = \Theta\left((\alpha^\star + 1)^2 L \log(1/\sigma_{\text{init}})/w^{\star 2}\right).$$

*Proof of Phase IV (convergence).*

By combining the loss (18), (21), and our parameter estimates (20), (29), it follows that:

$$\mathcal{L}_{\mathsf{G}_4}(\theta(t)) = \mathcal{O}\left(\frac{1}{t}\right), \quad \mathcal{L}_{\mathsf{IH}_2}(\theta(t)) = \mathcal{O}\left(\exp\left(-\Omega\left(\frac{w^{\star 2}t}{L(1+\alpha^\star)^2}\right)\right)\right), \quad t > T_{\text{III}}.$$

## E   USEFUL INEQUALITIES

**Lemma E.1** (Corollary A.7 in Edelman et al. (2022))**.** *For any $\theta, \theta' \in \mathbb{R}^d$, we have*

$$\|\text{softmax}(\theta) - \text{softmax}(\theta')\|_1 \leqslant 2\|\theta - \theta'\|_\infty$$

**Lemma E.2.** *For any $T \in \mathbb{N}_+$, $q, m \in \mathbb{N}_+$, there exist a $\phi_m^{\exp}(t) = \sum_{k=1}^m \alpha_k e^{-\beta_k t}$ such that*

$$\|\mathbb{I}(\cdot = T) - \phi_m^{\exp}(\cdot)\|_{\ell_1(\mathbb{N})} \leqslant C\frac{A^q(q^2)^q e^{0.01(q+1)T}}{m^q},$$

*where $\beta_k > 0$ holds for any $k \in [m]$, and $A, C > 0$ are absolute constants.*

*Proof of Lemma E.2.* This lemma is a corollary of Lemma F.1 in Wang and E (2024). By Lemma F.1 in Wang and E (2024) and its proof: for any $T \in \mathbb{N}_+$, $q, m \in \mathbb{N}_+$, there exists a $C(q) > 0$ and a $\phi_m^{\exp}(t) = \sum_{k=1}^m \alpha_k e^{-\beta_k t}$ such that

$$\|\mathbb{I}(\cdot = T) - \phi_m^{\exp}(\cdot)\|_{\ell_1(\mathbb{N})} \leqslant \frac{C(q)e^{0.01(q+1)T}}{m^q},$$

where $\beta_k > 0$ holds for any $k \in [m]$. Moreover,

$$C(q) = \frac{M(q)}{(1 - 1/e)^q}, \quad M(q) = \max_{0 \leqslant k \leqslant q} \sup_{x \in [-1,1]} \left|\Psi^{(k)}(x)\right|,$$

where $\Psi(x) = \begin{cases} \exp\left(-\frac{1}{1-x^2}\right), & x \in (-1, 1) \\ 0, & \text{otherwise} \end{cases}$ is the standard bump function on $[-1, 1]$. By a

straight-forward estimate, there exist absolute constants $C_1, C_2 > 0$ such that

$$\sup_{x \in [-1,1]} \left|\Psi^{(k)}(x)\right| \leqslant C_1(C_2)^k(k!)^2, \quad \forall k \in \mathbb{N}^+.$$

Thus, $C(q) \leqslant C_1 \left(\frac{C_2}{1-1/e}\right)^q (q!)^2$. By using Stirling's formula , there exists an absolute $C_3 > 0$ such that:

$$C(q) \leqslant C_1 \left(C_3 q^2\right)^q.$$

$\square$

**Lemma E.3.** $\underset{X,Y,Z}{\mathbb{E}} \exp(aXY)Z^2 = (1 - a^2)^{-1/2}, a < 1.$

*Proof of Lemma E.3.*

$$\int \exp(aXY)Z^2 \left(\frac{1}{2\pi}\right)^{-3/2} \exp(-\frac{1}{2}X^2 - \frac{1}{2}Y^2 - \frac{1}{2}Z^2) \, dX \, dY \, dZ$$

$$= \int \frac{1}{2\pi} \exp(-\frac{1}{2}(X - aY)^2 - \frac{1}{2}Y^2 + \frac{1}{2}a^2Y^2) \, d(X - aY)dY$$

$$= \int \frac{1}{\sqrt{2\pi}} \exp(-\frac{1}{2}W^2) \, dW \ (W = (1 - a^2)^{1/2}Y)$$

$$= (1 - a^2)^{-1/2}$$

$\square$

**Lemma E.4.** *Let* $M(p) := \frac{1 - e^{-p(L-2)}}{1 - e^{-p}}$, *then it holds that*

$$\left\|\mathrm{softmax}\left((-p(L - 1 - s))_{s=1}^{L-1}\right)\right\|_2^2 = \frac{M(2p)}{M(p)^2}.$$

**Definition E.5** (weakly majorizes). A vector $\mathbf{x} \in \mathbb{R}^n$ is said to *weakly majorize* another vector $\mathbf{y} \in \mathbb{R}^n$, denoted by $\mathbf{x} \prec_w \mathbf{y}$, if the following conditions hold:

1. $\sum_{i=1}^k x_{[i]} \leqslant \sum_{i=1}^k y_{[i]}$ for all $k = 1, 2, \ldots, n - 1$,

2. $\sum_{i=1}^n x_{[i]} = \sum_{i=1}^n y_{[i]}$,

where $x_{[i]}$ and $y_{[i]}$ are the components of $\mathbf{x}$ and $\mathbf{y}$, respectively, arranged in decreasing order.

**Lemma E.6** (Weighted Karamata Inequality). *Let* $f : \mathbb{R} \to \mathbb{R}$ *be a convex function, and let* $\mathbf{x} = (x_1, x_2, \ldots, x_n)$ *and* $\mathbf{y} = (y_1, y_2, \ldots, y_n)$ *be two vectors in* $\mathbb{R}^n$. *If* $\mathbf{x}$ *weakly majorizes* $\mathbf{y}$ *(i.e.,* $\mathbf{x} \prec_w \mathbf{y}$), *and* $w_1, w_2, \ldots, w_n$ *are non-negative weights such that*

$$\sum_{i=1}^n w_i = 1,$$

*then the following inequality holds:*

$$\sum_{i=1}^n w_i f(x_i) \leqslant \sum_{i=1}^n w_i f(y_i).$$

# F   LARGE LANGUAGE MODELS USAGE STATEMENT

In adherence to the ICLR 2026 policy, we disclose the use of a large language model (LLM) as a general-purpose writing assistant during the preparation of this manuscript. The LLM's role was strictly limited to improving the clarity, grammar, and readability of our author-written text, such as spell-checking and rephrasing sentences for better flow. Crucially, the LLM did not contribute to any of the core scientific aspects of this work, including research ideation, experimental design, data analysis, or the generation of novel insights. The authors have carefully reviewed all LLM-modified text and take full responsibility for the intellectual substance and final content of this paper.

