# OpenReview forum: "How Transformers Get Rich: Approximation and Dynamics Analysis"
_ICLR.cc/2026/Conference — ICLR 2026 Conference Withdrawn Submission_

### Official Review · Reviewer_GK55 · 2025-10-30

**Soundness:** 3
**Presentation:** 3
**Contribution:** 3
**Rating:** 6
**Confidence:** 3

**Summary:**

This paper provides a theoretical analysis of the in-context learning capabilities of a transformer based on the understanding of induction heads. The paper proposes an approximation analysis and a dynamics analysis. In approximation analysis, it is shown that in addition to prior works showing vanilla induction heads, alternate layer combinations of transformers can also achieve to form induction heads, e.g., a 2 layered attention network without FFNs is shown to be able to construct an induction head. For dynamics analysis, the paper considers a target function consisting of a 2-gram and a 4-gram component and trains it in 2-stage showing that in the first stage the model learns the 2-gram, whereas in the second stage it transitions to the induction head learning learning.

**Strengths:**

- The paper provides a theoretical understanding of the working of induction heads in transformers. First, in approximation analysis, it shows that a transformer with 2 attention layers without FFNs can achieve induction heads by proving it by construction.
- For analyzing the training dynamics, the paper proposes a target function consisting of 2-gram and 4-gram components, then a layerwise training is done to show that the model first learns the induction head, followed by the second stage of training where the model transitions to the 4-gram.
- Some experimental validations are provided for the approximation as well as dynamics analysis.

**Weaknesses:**

- The results in the approximation analysis is one way of constructing and explaining the working of transformers. Can the authors provide further evidence if the constructions used in the proofs are indeed how the 2-layered transformers work? E.g., moving from Thm3.3 to Thm3.4, in line 277, it is discussed that the FFN is used for approximation, which is intuitively correct, but is there a way to check if this actually is what is happening in the transformer? There are several other such constructions used in the paper without proper theoretical or experimental justifications.
- The setup for the dynamics analysis seems to restrictive: both the dataset and the model setup is very restrictive, hence, even though the observations of emergence and analysis of how the parameters vary during the training is very interesting, it is not clear whether this helps explain the training dynamics of larger transformers
- The writing can be improved. Since the paper contains a lot of mathematical notations, it would help to provide overview of the construction and theorem to follow before providing all the notations.

**Questions:**

Please check the weaknesses if they can be addressed

---

### Official Review · Reviewer_LuGK · 2025-10-31

**Soundness:** 2
**Presentation:** 3
**Contribution:** 2
**Rating:** 4
**Confidence:** 4

**Summary:**

The paper studies how small transformers acquire induction heads — mechanisms that retrieve a repeated pattern in context (e.g. see ... a b ... a ⇒ predict b). It makes two contributions:

1. Expressivity / approximation. The authors formalize several induction-style mechanisms and prove approximation results for single/multiple head attention w/o FFNs. These constructions assign concrete algorithmic roles to architectural pieces: relative positional encoding to align candidate matches; attention heads to retrieve the continuation; multiple heads to store multi-token patches; FFNs to learn richer similarity.

2. Training dynamics / phase transition.
The paper studies a 2-layer transformer trained on a mixed objective combining a simple local $n$-gram rule and an induction-style copying rule. They show analytically (and confirm experimentally on synthetic data and natural text) that training proceeds in phases: the model first learns the easy local heuristic, then hits a plateau, and only later “snaps” into using an induction head. The authors explain this as a time-scale separation: the $n$-gram rule depends on parameters that learn quickly (essentially linear), while induction-style retrieval depends on slower-growing quadratic interactions of attention weights.

The main claimed contribution is a unified theory linking (a) which minimal transformer components are sufficient to implement induction heads, and (b) why induction heads emerge suddenly during training, rather than gradually.

**Strengths:**

Though I did not carefully check the proof, the technical details seem sound to me. The presentation of the paper is clear and logical. The work also proposes a clean, analyzable setting, and the experimental evidence is aligned with the theory.

The paper gives a unified conceptual bridge between two areas that are usually disconnected: (i) mechanistic accounts of “induction heads” and (ii) the actual temporal trajectory of training under gradient-based optimization. Even if individual ingredients (like induction heads themselves, or kernel-like FFN constructions) have appeared before, treating the emergence of an induction head as a phase transition with identifiable stages and drivers is a distinctive narrative contribution.

The paper also proposes a clean hierarchy of induction-style mechanisms ($\mathrm{IH}_2 \rightarrow \mathrm{IH}_n \rightarrow \mathrm{GIH}_n$) and ties each step to specific architectural components of a 2-layer transformer (RPE, multi-head attention, FFN). That systematic decomposition feels like a useful reframing, not just a re-statement of known folklore.

**Weaknesses:**

However, I have some major concerns about the current work:

**1. Realism / motivation of the mixed target**

The core dynamics result is proved in a very specific “mixed target’’ setting where the ground truth is a convex combination of (i) a handpicked 4-gram rule and (ii) a vanilla 2-gram induction-head-style copying rule. It’s not obvious when this exact mixture arises in real next-token prediction. The paper justifies 4-gram instead of 2/3-gram mainly to avoid trivial cases where the model can already recover the last few tokens from residual connections. But this is still a very controlled synthetic target rather than a distributional property of natural text, and I cannot see what brings up special difficulty in predicting $x_{L-2}$ rather than $x_{L-1}$.

Because the theory depends on that mixture — in particular, on the relative weights of the two components ($\alpha^\star$) and their very different gradient magnitudes, which drive the time-scale separation and plateau/“snap’’ behavior — it’s unclear how strongly the claimed four-phase training story transfers beyond this stylized setup. The paper does include experiments on WikiText-2 showing qualitatively similar plateaus and norm growth, which is reassuring, but these are observational rather than guaranteed by the theory.

As a reviewer, I would expect a clearer argument that this mixed-target analysis is more than an illustrative toy: e.g., is there a natural linguistic or algorithmic setting where both a short-range heuristic and an induction-style retrieval signal are jointly “in the labels’’ in the way assumed here?

**2. Scope / simplifications of the theoretical model**

All formal dynamics results are for a depth-2 transformer without FFNs, under gradient flow (continuous-time GD), on 1D or very low-dimensional synthetic sequences, with small initialization and long-context asymptotics. This is standard for theoretical work, but it raises external validity concerns: modern LLMs use many layers, FFNs, layernorm, Adam, etc. The paper does show that Adam in higher dimension qualitatively exhibits multiple plateaus, which matches the story, but the analysis itself does not cover Adam or deep stacks.

Also, I believe the authors should also explain in more details what is the purpose of separating the training stages, and why the separation of the stages will not influence the conclusion. In short, the phenomena are compelling, but the formal guarantees are still quite far from the full training regime people care about.

**3. Clarity of novelty vs prior work on induction heads.**
The approximation results are positioned as identifying which architectural pieces (multi-head attention, RPE, FFNs) are sufficient to implement increasingly general induction-head-like mechanisms: vanilla $\mathrm{IH}_2$, $n$-gram-style $\mathrm{IH}_n$, and a generalized similarity-based $\mathrm{GIH}_n$. However, prior work has already shown that shallow transformers can implement induction-head behaviors and even converge to them in training (e.g., Nichani et al., 2024; Chen et al., 2024b, as cited in the paper).

The paper argues its contribution is (i) tighter sufficiency for two-layer transformers with multi-head attention (rather than deeper/specially wired transformers), and (ii) explicit approximation rates and head-count requirements, e.g. needing $H = \Omega(n)$ heads to express $\mathrm{IH}_n$ well. But the exposition doesn’t fully spell out how this improves on recent variants that also express induction-like copying using kernel-style feature maps or FFN-learned bases. In particular, the generalized induction head $\mathrm{GIH}_n$ is implemented by having the FFN approximate a set of basis functions (via something like a POD / low-rank decomposition of a similarity kernel $g$), and then using attention to recombine them. This sounds close in spirit to existing “FFN-as-feature-map” or “FFN as dictionary / kernel expansion” views in the literature, where the FFN builds features that attention then uses to perform matching. The paper cites related lines but doesn’t precisely articulate what is strictly new versus what is a cleaned-up restatement in their notation. Therefore, it remains hard to evaluate the novelty of the results.

**4. Baselines and comparison to related dynamics studies.**
The paper positions its dynamics analysis as going beyond prior empirical observations of “snap-to-induction,” claiming a full four-phase theoretical account and a mechanistic explanation (time-scale separation + component weighting). But there’s limited comparison to other recent theoretical studies of attention learning dynamics (e.g., Huang et al., Chen et al., Eshaan et al. mentioned in the submission). The related work section acknowledges them, but the body of the paper doesn’t clearly isolate what technical obstacles those works couldn’t resolve that this paper now resolves.

**Questions:**

See the weakness session

---

### Official Review · Reviewer_ZnJ2 · 2025-10-31

**Soundness:** 3
**Presentation:** 3
**Contribution:** 2
**Rating:** 6
**Confidence:** 4

**Summary:**

This paper studies how simplified transformers can implement induction heads and analyzes the learning process from the perspective of training dynamics. For approximations, the paper proves that there exist transformers that can implement vanilla and more generalized induction heads. For dynamics, this paper demonstrates the model's transition from a 4-gram pattern to the induction head mechanism.

**Strengths:**

1. The induction head is an interesting and important mechanism in transformer research, and this paper constructs a comprehensive theoretical framework for it. The modeling of the induction head is intuitively reasonable. Its progressive analysis is logical and supported by rigorous theoretical proof.
2. Many proofs in transformer theory research involve artificially constructing the model's weights for subsequent analysis. While the first part of this paper also utilizes this technique, the study of training dynamics enhances the practical significance of this paper.
3. The authors conduct extensive experiments to support their theoretical results.

**Weaknesses:**

1. I think the analysis of the approximation of induction head seems to have appeared in previous work, such as [1]. The difference between this work and previous papers may be that it studies the training dynamics from 4-gram to the induction head. This may weaken the contribution of this paper.
2. This paper demonstrates in the approximation part that the transformer can achieve induction heads by constructing parameters. Although there is an analysis of the training dynamics later, this does not seem to demonstrate the learnability of the constructed parameters, that is, whether the model can truly learn the parameters constructed in the approximation part.

[1] Nichani, Eshaan, Alex Damian, and Jason D. Lee. "How transformers learn causal structure with gradient descent."

**Questions:**

Can you discuss the main differences and contributions of this paper compared to previous similar works?
Can you provide some explanation for the learnability?

---

### Official Review · Reviewer_Crcx · 2025-11-05

**Soundness:** 2
**Presentation:** 4
**Contribution:** 2
**Rating:** 2
**Confidence:** 4

**Summary:**

This paper investigates the dynamics behind the formation of induction heads in Transformers.Initially, the authors propose a formal definition for an induction head and introduce a generalized version. The paper then quantitatively studies how induction heads are implemented within two-layer Transformers, analyzing architectures both with and without MLP layers.In the latter part of the study, the authors define a stylized task: learning a mixture of a $4$-gram and an in-context $2$-gram model. They analyze the learning dynamics of this task using a layer-wise gradient flow.Their analysis reveals that while training the second layer, the learning processes for the $4$-gram and the in-context $2$-gram are decoupled. Moreover, they show that the formation of the induction head (corresponding to the $2$-gram task) happens in a phase-wise manner for small initialization.

**Strengths:**

Given that induction heads are widely assumed to be critical for in-context learning, their formation dynamics have become a focal point of recent research. This paper contributes to this area by establishing grounding definitions for induction heads and studying how these mechanisms are represented within the Transformer architecture.
The proposed simplified model architectures and a specialized task isolate and analyze their formation better.

**Weaknesses:**

**Validity of Theorems:** There appears to be an issue with Theorems 3.3 and 3.4 (a specific question regarding this is detailed below)

**Mischaracterization of "Lazy" Learning** I disagree with the paper's description of the task dynamics as a "lazy" phenomenon. The learning of $f_{G_4}^*$ is still a form of feature learning and does not align with the formal definitions of "lazy" (or kernel-regime) learning established in prior work (e.g., Chizat et al., 2018; Woodworth et al., 2020).

**Dependence on Positional Encoding:** The paper notes that the observed timescale separation is due to the quadratic dependence on $w_{KQ}$ versus the linear dependence on the Relative Positional Encoding (RPE). This suggests that the separation in learning speeds would vanish if a different positional encoding were used(for e.g., adding positional embedding to token embedding would recover the quadratic dependence). Therefore, it is problematic to frame this as a general dynamics transitioning from $n$-gram to induction head learning; it seems to be an artifact of the specific RPE implementation rather than a fundamental dynamic.

**Questions:**

**Problem with the proof of Theorem 3.3 and Theorem 3.4**:  I find something missing in the the proofs and I would like the authors to clarify it.

In the step 1 of proof of Theorem C.2 when mapping from $(x_s, 0) \to \( x_s, x_{s-1}, \ldots, x_{L-n+1} \)$,  particularly in lines 1107 to 1100 there is problem with normalization coefficients as described below.
$$\begin{align*}
SA^{1,h}(X\_{0:s}^0)\_{-1} = \sum\_{j} a^{h}\_{s-j,s}  W_V^{1,h} x_{s-j} = S_i \\alpha\_{h,i} \\left( \\sum\_{f=0}^{H\_i} \\exp{  \\left( -\\beta\_{h,i} ( f - 1) \\right) }  \\right) \sum\_{j} a^{h}\_{s-j,s} x_{s-j}
\end{align*}$$ where $a\_{s-j,s}$ is the attention score and $a\_{s-j,s} = \frac{e^{-\beta\_{h,i} (j- 1)} }{ \\sum\_{f = 1}^{s} e^{-\\beta\_{h,i} (f- 1)} }$. So,
$$\begin{align*}
SA^{1,h}(X\_{0:s}^0)\_{-1} =  S_i \\alpha\_{h,i} \frac{\\left( \\sum\_{f=0}^{H\_i} \\exp{  \\left( -\\beta\_{h,i} ( f - 1) \\right) }  \\right)}{ \\sum\_{f = 1}^{s} e^{-\\beta\_{h,i} (f- 1)} } \sum\_{j} e^{-\beta\_{h,i} (j- 1)} x_{s-j}
\end{align*}$$ Note that the normalization here does not cancel out and adding it from head $h = \\sum\_{j=1}^{i-1} H\_j , \ldots,  \\sum\_{j=1}^{i} H\_j$ does not recover the left hand side of equation in lines 1107- 1110. Even a different choice of value matrix does not fix it as it should not be dependent on $s$ and should work for any length. If this confusion is clarified I would reconsider my evaluation.

---

### Note · Authors · 2026-01-05

**Comment:**

We would like to thank all the reviewers for the detailed and constructive feedback. We learned a lot from the review and will update our manuscript accordingly in the future.

**Withdrawal Confirmation:**

I have read and agree with the venue's withdrawal policy on behalf of myself and my co-authors.